TOPICAL REVIEW

# Compromised cardiopulmonary transition in fetal growth restricted and small for gestational age neonates

Zahrah Azman[1,2] , Arvind Sehgal[3,4] , Suzanne L. Miller[1,2] , Kristen J. Bubb[5,6] , Graeme R. Polglase[1,4] and Beth J. Allison[1,2]

[1] *The Ritchie Centre, Hudson Institute of Medical Research, Clayton, Victoria, Australia*
[2] *Department of Obstetrics and Gynaecology, Monash University, Clayton, Victoria, Australia*
[3] *Monash Newborn, Monash Medical Centre, Clayton, Victoria, Australia*
[4] *Department of Paediatrics, Monash University, Clayton, Victoria, Australia*
[5] *Biomedicine Discovery Institute, Monash University, Clayton, Victoria, Australia*
[6] *Victorian Heart Institute, Monash University, Clayton, Victoria, Australia*

Handling Editors: Laura Bennet & Janna Morrison

The peer review history is available in the Supporting Information section of this article (https://doi.org/10.1113/JP289441#support-information-section).

The Journal of Physiology

G. R. Polglase and B. J. Allison contributed equally to this work.

The Journal of Physiology

**Abstract figure legend** Differences in cardiovascular function between appropriately grown (AG) and fetal growth restricted (FGR) infants after the cardiopulmonary transition at birth. AG infants typically exhibit an increase in left ventricular output (LVO) and stroke volume (SV) in the first few days after birth, along with a reduction in myocardial performance index (MPI) and pulmonary vascular resistance (PVR). Together, these normal longitudinal changes in cardiac function facilitate more efficient ventricular function and improved pulmonary blood flow to meet the increasing demands of the postnatal circulation. In contrast, FGR and SGA (small for gestational age) infants often fail to show these adaptations, indicating a compromised cardiopulmonary transition and increased susceptibility to additional antenatal stressors. SGA data are included given the historical misclassification of FGR prior to the Delphi consensus and the limited availability of neonatal cardiovascular data in strictly defined FGR populations.

**Abstract** The cardiopulmonary transition at birth is a critical physiological process requiring coordinated cardiovascular adaptation to meet the increased circulatory demands of extrauterine life. This transition may be compromised in infants affected by suboptimal fetal growth, such as in infants born small for gestational age (SGA) or classified with fetal growth restriction (FGR). Suboptimal fetal growth often arises from reduced oxygen and nutrient supply, leading to prioritised perfusion of crucial organs and subsequent cardiac and arterial remodelling. These cardiovascular adaptations, while necessary for fetal survival, may persist postnatally and increase the risk of an impaired cardiovascular transition at birth. Altered echocardiographic function and cardiac injury biomarkers are often detectable in this population during the early postnatal period, indicating underlying myocardial stress and a predisposition to an impaired transition. FGR and/or SGA neonates often exhibit impaired diastolic function, reflecting impaired myocardial relaxation and reduced compliance, and systolic dysfunction, including a reduced capacity to increase left ventricular output over time. Additionally, elevated pulmonary vascular resistance contributes to an increased risk of respiratory morbidity. Emerging preclinical data suggest that these adaptations may impede the neonate's ability to respond to perinatal stressors, thus increasing the risk of adverse outcomes. Understanding the multifaceted nature of cardiovascular dysfunction in FGR and/or SGA infants during the perinatal period is essential to improving their long-term outcomes, thus reducing the risk of cardiovascular disease later in life.

(Received 16 June 2025; accepted after revision 21 August 2025; first published online 15 September 2025)

**Corresponding author** Zahrah Azman and Beth J. Allison: The Ritchie Centre, Hudson Institute of Medical Research, Clayton, VIC, Australia. Email: siti.azman@monash.edu; beth.allison@hudson.org.au

## Introduction

Cardiovascular disease is the leading cause of mortality globally, accounting for 32% of all deaths worldwide (World Health Organization, 2021). Epidemiological studies have long established an association between perturbations to fetal development with an increased risk of cardiovascular disease in adulthood (Barker, Osmond et al., 1989; Barker, Osmond, Winter et al., 1989; Barker, 1990). Conditions of impaired fetal growth are also strongly linked to cardiovascular dysfunction in the neonatal period (Crispi et al., 2018). Infants born small for gestational age (SGA) are defined by an estimated fetal weight below the 10th percentile for gestational age, whereas fetal growth restriction (FGR) additionally requires the confirmation of abnormal Doppler findings indicative of placental pathology, such as increased pulsatility index or absent/reversed end-diastolic flow in

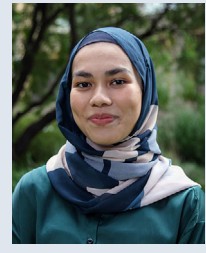

**Zahrah Azman** is a PhD candidate at The Ritchie Centre, Hudson Institute of Medical Research and Monash University. Her PhD investigates how fetal growth restriction (FGR), a pregnancy complication arising from placental insufficiency, impacts neonatal cardiovascular function. Her research employs *in vivo*, *ex vivo* and molecular techniques to investigate cardiac function and energy utilisation during the perinatal period. In the future, she is interested in understanding how cardiovascular complications evolve across the lifespan in at-risk populations, with the goal of identifying therapeutic strategies to mitigate the risk of long-term cardiovascular disease.

the umbilical artery (Figueras & Gratacós, 2014; Lees et al., 2020). Although a proportion of SGA infants may be constitutionally small, it is important to recognise that up to 50% of FGR cases remain undetected before birth, resulting in a population of SGA infants potentially having an unrecognised or subclinical pathology (Ernst et al., 2017; Lappen & Myers, 2017). Thus, it is critical that FGR and SGA infants are considered separately, given that the former is representative of a pathological condition and the latter reflects constitutionally small individuals. Further, given the current low rate of detection of true FGR, the ability to accurately distinguish between FGR and SGA populations remains a clinical challenge, underscoring the need for mechanistically informed biomarkers, as will be discussed later, that may have the potential to add clarity to clinical interpretation of Doppler scans.

The increased risk of cardiovascular disease likely results from the fetal adaptive responses to an adverse intrauterine environment that permanently alters cardiovascular development, thus increasing the susceptibility to postnatal cardiovascular injury (Armengaud et al., 2021). The heart is central to the fetal adaptive response to intrauterine insults (e.g. toxin exposure, twin-twin transfusion syndrome, preeclampsia and FGR) and remodels in response to changes in preload and afterload to maintain optimal oxygenation of critical developing organs such as the brain (García-Otero et al., 2016; Karatza et al., 2002; Rodríguez-López et al., 2017; Rychik et al., 2007; Youssef et al., 2020). Unsurprisingly, fetal cardiac morphological adaptations are often accompanied by changes in function, predisposing the neonate to cardiac dysfunction (Crispi et al., 2014; Pérez-Cruz et al., 2015; Rodríguez-López et al., 2017). Cardiac remodelling can also result in a reduced ability to tolerate physiological challenges to the cardiovascular system. The fetus must also encounter one of the greatest cardiovascular challenges in its lifetime: the cardiopulmonary transition at birth, which, if unsuccessful, increases the risk of circulatory failure during the early adaptation to postnatal life. This review will include studies describing both FGR and SGA and examine perinatal cardiovascular outcomes related to each condition.

## Cardiovascular adaptations to placental insufficiency

The primary cause of impaired fetal growth is placental insufficiency, although other maternal, fetal and genetic factors can contribute to its pathogenesis (Burton & Jauniaux, 2018). Placental insufficiency reduces the amount and efficiency of blood flow from the placenta to the fetus, resulting in decreased fetal oxygen and nutrient transfer and subsequent chronic fetal hypoxaemia (Zhang et al., 2015). The concept of fetal programming, first

identified by David Barker, highlights that the fetal adaptive response to an unfavourable intrauterine environment initially ensures survival and development but results in changes that may persist into postnatal life (Barker, Osmond et al., 1989; Barker, 1990; Barker et al., 2009). Importantly, the long-term impact of placental insufficiency on the fetus is strongly associated with an increased risk of cardiovascular disease later in life (Arnold et al., 2015; Leon et al., 1998). The fetal cardiovascular system activates compensatory mechanisms in response to acute hypoxia resultant from placental insufficiency, termed a 'brain sparing response' (Miller et al., 2016). The brain sparing response occurs in an attempt to maintain oxygen delivery to vital organs by altering the perfusion of various vascular beds, therefore increasing or maintaining blood flow to critical organs, such as the brain and adrenal glands (Giussani, 2016). This occurs at the expense of less critical organs, including the lungs and gastrointestinal tract (Giussani, 2016). While FGR is associated with the brain sparing response, this does not consistently translate to an increase in overall cerebral blood flow (Darby et al., 2024; Miller et al., 2009; Poudel et al., 2015; Zhu et al., 2016). Instead, the redistribution of regional cerebral perfusion may explain the elevated middle cerebral artery blood flow velocities observed clinically (Eixarch et al., 2008). Additionally, although blood flow is preferentially redistributed to critical organs following chronic hypoxaemia, overall substrate delivery is reduced compared to normal pregnancies, resulting in reduced fetal substrate consumption relative to oxygen delivery (Cetin et al., 2020).

Chronic hypoxia, defined as exposure to reduced oxygen content for weeks to months, involves the persistence of the circulatory brain sparing response and maintenance of substrate redistribution towards essential circulations (Allison, Brain et al., 2016). Despite this, the term 'brain sparing' may be misleading as clinical evidence regarding its role in neurodevelopmental injury (von Beckerath et al., 2013; Morsing et al., 2021) or protection (Malhotra et al., 2015; Piscopo et al., 2025) in FGR infants remains contradictory. The fetal brain sparing response to acute hypoxia is activated by carotid chemoreflexes acting on the brainstem, although the exact contribution of the brainstem in maintaining the response to chronic hypoxaemia remains unclear (Giussani, 2016). Preclinical studies have investigated the effects of chronic hypoxaemia on the brainstem, reporting conflicting outcomes including minimal structural impact (Adams et al., 2001; Thordstein & Hedner, 1992; Tolcos et al., 2003), reduced brainstem blood flow (Miller et al., 2009) and dysregulated monoaminergic and neuroglial development (Ahmadzadeh, Dudink et al., 2023; Tolcos & Rees, 1997). Regardless, it is possible that neuropathology in the brainstem contributes to the poor cardiorespiratory outcomes of FGR infants after birth

(Ahmadzadeh, Polglase et al., 2023). Chronic brain sparing, maintained in the presence of hypoxaemia, is also known to cause maladaptive morphological adaptations to the heart, resulting in cardiac remodelling (Pérez-Cruz et al., 2015; Rodríguez-López et al., 2017), disrupted cardiomyocyte maturation (Bubb et al., 2007; Jonker et al., 2018; Louey et al., 2007; Morrison et al., 2007) and vascular dysfunction (Bubb et al., 2007; Sehgal et al., 2013; Tare et al., 2014; Thompson, Gros et al., 2011; Thompson et al., 2014), ultimately compromising cardiovascular function. Importantly, the timing, duration and severity of the underlying causes of FGR each uniquely influence the developmental trajectory of the heart and other organs, leading to variable structural and functional outcomes (Darby et al., 2020; Dudink et al., 2025; Morrison, 2008). These *in utero* adaptations often reflect underlying placental pathology, as reflected in a study of very preterm FGR infants noting greater maternal/fetal malperfusion compared to equally premature but appropriately grown (AG) infants (Sehgal,

Dahlstrom et al., 2019). Placental histopathology and associated biomarkers therefore provide a better understanding into the pathophysiology of perinatal cardiovascular disease in this vulnerable population.

**Fetal cardiac remodelling.** Peripheral vasoconstriction is a key driver of the brain sparing response and, together with elevated placental vascular resistance, ultimately results in an increase in afterload (Giussani, 2016). This gives rise to pressure- or volume-induced cardiac remodelling (Crispi et al., 2018). In the developed adult heart, mild chronic pressure overload (as seen in aortic stenosis) leads to concentric left ventricular (LV) hypertrophy in the most stressed regions through the reorganisation of sarcomere structures within cardiomyocytes without significant changes in chamber size, thereby maintaining active cardiac mechanics (Fig. 1) (Pitoulis & Terracciano, 2020). The Frank-Starling mechanism, which highlights an increase in cardiac

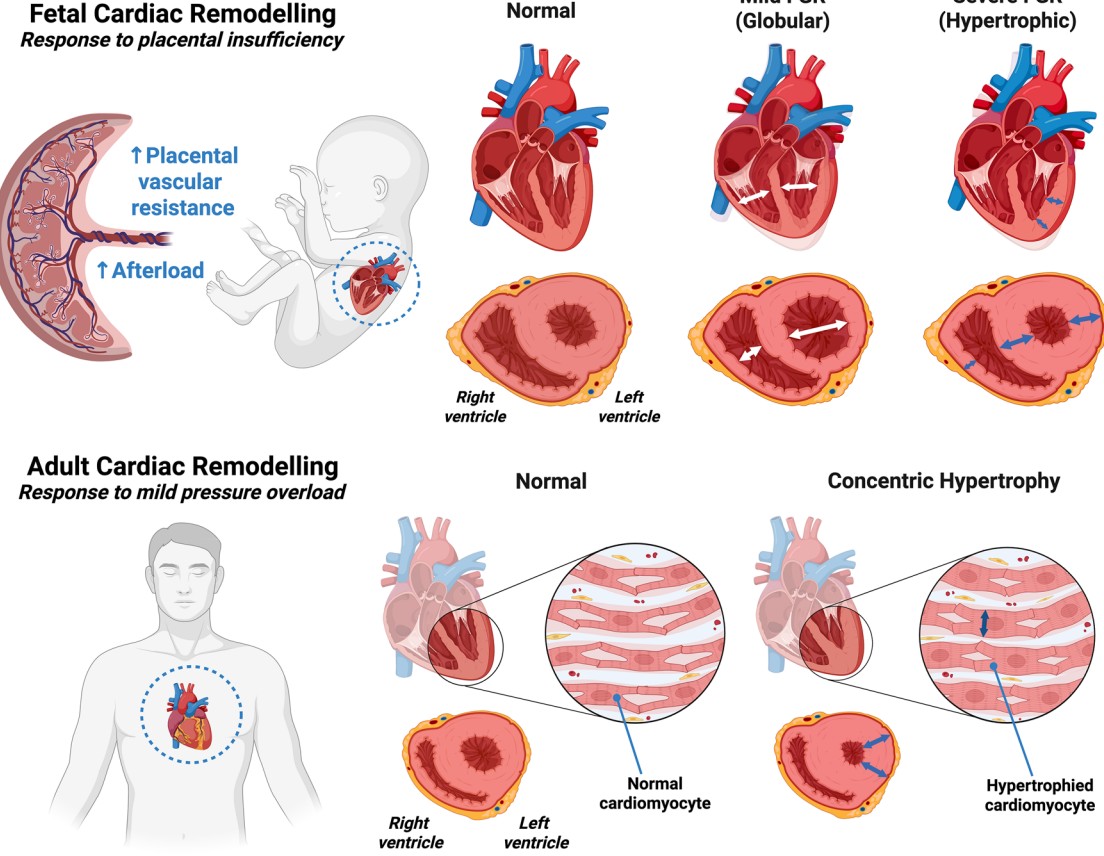

**Figure 1. Fetal *vs.* adult cardiac remodelling in response to pressure overload**
In fetal growth restriction (FGR), placental insufficiency chronically increases placental vascular resistance and afterload. Unlike adults, fetal cardiomyocytes have a limited capacity for concentric hypertrophy, leading to distinct remodelling phenotypes. In mild FGR, a globular ventricular shape develops to maintain cardiac output with less contraction force while reducing wall stress. Concentric hypertrophy only arises later with increasing severity of placental insufficiency (severe FGR). In adults, mature cardiomyocytes adapt to mild pressure overload through concentric hypertrophy via sarcomeric addition in parallel, increasing wall thickness without altering chamber size.

output (CO) per unit increase in preload, is enacted in circumstances of mild overload (Pitoulis & Terracciano, 2020). However, the Frank-Starling mechanism is less functional during fetal life, with the fetal heart having a limited ability to adjust CO in response to increased preload (Gilbert, 1980; van Hare et al., 1990; Teitel et al., 1991).

The limited ability of the fetal heart to adjust to pressure and volume overload results in unique remodelling of the growth-restricted heart (Fig. 1) (Crispi et al., 2020; Rock et al., 2023; Youssef et al., 2023). Pressure overload, which is more commonly implicated in the remodelling of the growth-restricted heart, is driven by an increase in placental vascular resistance. This increases the resistance to ventricular ejection during systole (after-load) (Crispi et al., 2020). The growth-restricted heart may also experience volume overload, as concurrent vaso-dilatation of the cerebral arteries results in a decrease in ventricular afterload (Verburg et al., 2008), resulting in improved right ventricular (RV) function without fibrosis or apoptosis (Karamlou et al., 2019; Toischer et al., 2010). Exposure to volume/pressure overload initially results in dilation of the RV (Verburg et al., 2008) that forces the septum toward the LV, resulting in a globular RV and elongated LV (Crispi et al., 2018, 2020). This 'elongated' phenotype allows the heart to tolerate volume overload with a reduction in wall stress and can affect almost a third of the fetal FGR population (Rodríguez-López et al., 2017). The mechanistic pathways contributing to cardio-vascular sequelae in FGR may also offer a foundation for developing biomarkers to improve the diagnosis of FGR.

Increased duration or severity of placental insufficiency leads to a sustained increase in cardiac afterload throughout gestation, resulting in severe cardiac remodelling (Crispi et al., 2018). Sustained pressure overload produces more spherical right and left ventricles and a 'globular' phenotype, allowing the heart to maintain CO with less contraction force while reducing wall stress (Fig. 1) (Crispi et al., 2018, 2020). On echocardiographic assessment, this spherical appearance reflects in a lower sphericity index in the postnatal period in preterm FGR infants (Sehgal et al., 2017). Despite maintaining CO, increased ventricular globularity in growth-restricted fetuses results in a reduction in stroke volume (SV), thereby requiring an increase in heart rate to match CO (Crispi et al., 2010). A study by Rodriguez-Lopez et al. determined that 54% of SGA cases present with a globular cardiac phenotype (Rodríguez-López et al., 2017). Elongated and globular cardiac phenotypes were associated with less severe SGA cases, which were often classified as late- rather than early-onset (Rodríguez-López et al., 2017). Additionally, fetuses with a globular-shaped RV (i.e. elongated phenotype) had a lower risk of perinatal death compared to fetuses with increased globularity in both ventricles (i.e. globular

phenotype) (DeVore et al., 2021). With increasing severity or prolonged duration of pressure overload, an increase in sphericity is no longer sufficient to tolerate wall stress, resulting in a hypertrophic phenotype characterised by myocardial hypertrophy and cardiomegaly that is strongly associated with severe, early-onset growth restriction (Rodríguez-López et al., 2017). Importantly, this fetal ventricular hypertrophy is caused by structural remodelling rather than true hypertrophic growth at the cardiomyocyte level (Fig. 1).

Hypertrophic thickening of the ventricular walls reduces the space within the ventricle, resulting in a less dilated ventricular cavity with a smaller inner diameter, allowing the generation of higher force and pressure with less energy consumption and without additional stress to individual cardiomyocytes (Crispi et al., 2020). However, hypertrophy of the ventricles ultimately impairs cardiac function, resulting in decreased longitudinal function and impaired relaxation (Crispi et al., 2020). Myocardial hypertrophy is not limited only to the ventricular walls and can also manifest in the intraventricular septum, leading to LV outflow obstruction (Valenzuela-Alcaraz et al., 2019).

**Clinical studies on cardiac adaptation in growth-restricted fetuses.** Numerous studies of FGR and/or SGA fetuses have linked cardiac remodelling with functional impairment of the heart (Table 1). A prospective cohort study of fetuses in the final trimester of gestation described late-onset FGR fetuses as having systolic dysfunction, as evidenced by decreased tricuspid annular plane systolic excursion (TAPSE) and LV myocardial performance index (MPI) (Pérez-Cruz et al., 2015). Markers of diastolic dysfunction were also observed, including abnormal mitral and tricuspid early (E) to late (A) ventricular filling velocities during diastole (E/A ratios), indicating impaired LV and RV relaxation, respectively (Pérez-Cruz et al., 2015). Similarly, FGR fetuses examined at 24–34 weeks' gestation had a decrease in left, but not right MPI, suggesting global LV dysfunction (Comas et al., 2010). Additional Doppler imaging demonstrated lower systolic and diastolic myo-cardial velocities in the mitral and tricuspid annuli, suggesting impaired myocardial contraction (Comas et al., 2010). In addition to impairments in myo-cardial motion, dysfunction in myocardial deformation was also identified in FGR fetuses examined by 2D speckle-tracking echocardiography where lower global and segmental LV strain values were observed in FGR fetuses (Domínguez-Gallardo et al., 2023). Crispi et al. observed biphasic segmental shortening in SGA fetuses, with initial systolic deformation reaching end-systolic maximum strain followed by additional post-systolic shortening during isovolumetric relaxation, compared

**Table 1. Studies of FGR and/or SGA cardiovascular structure and function during gestation**

| Author (year) | Definition of FGR and/or SGA | Gestational age at study | Number of fetuses | Echocardiographic outcomes |
|---|---|---|---|---|
| Comas et al. (2010) | Early-onset FGR: EFW <10th centile + UA PI >95th centile | 24–34 weeks | FGR $n = 25$ <br> AG $n = 50$ | FGR had increased left MPI but similar E/A ratios, outflow tract velocities and right MPI. FGR had decreased systolic and diastolic tricuspid and mitral myocardial velocities, increased mitral E'/A', increased mitral, tricuspid and septal MPI |
| Crispi et al. (2014) | SGA: EFW and BW <10th centile | ~32 weeks | SGA $n = 37$ <br> AG $n = 37$ | SGA fetuses had more globular hearts, reduced systolic motion and prolonged IVRT. Over half of SGA fetuses exhibited abnormal postsystolic shortening, which was associated with absent hypertrophy, poorer perinatal outcomes, and higher postnatal blood pressure. |
| DeVore et al. (2021) | Early-onset FGR (<34 weeks' GA): EFW <10th centile + absent/reversed end-diastolic velocity of the UA | <34 weeks | FGR $n = 49$ (Neonatal survivors $n = 36$; perinatal deaths $n = 13$) | Neonatal survivors had decreased RV sphericity index, decreased LV/RV areas, decreased LV/RV widths and increased RV apical transverse widths compared to perinatal deaths |
| Domínguez-Gallardo et al. (2023) | SGA: EFW <10th centile | ~32 weeks | SGA $n = 45$ <br> AG $n = 137$ | SGA fetuses had lower LV global and segmental strain values compared to AG fetuses |
| Oluklu et al. (2023) | SGA: EFW <10th centile classified into early-onset (<32 weeks) and late-onset (>32 weeks) | Early-onset: 32–33 weeks <br> Late-onset: 36–37 weeks | SGA $n = 82$ (Early-onset $n = 28$; late-onset $n = 54$) <br> AG $n = 82$ | Both early- and late-onset SGA had decreased RV/LV sphericity index. Early-onset SGA had increased RV wall thickness compared to AG, while late-onset SGA had increased LV wall thickness compared to AG. Both early- and late-onset exhibited systolic and diastolic impairments including increased TAPSE, MAPSE, MPI and ejection time. Early-onset had more pronounced changes in myocardial velocity, including increased LV and RV s' velocities, tricuspid and mitral A', and mitral E'/A' |
| Pérez-Cruz et al. (2015) | SGA: EFW between 3rd and 9th centile <br> FGR: EFW <3rd centile or <10th centile + cerebroplacental ratio <5th centile and/or UA PI >95th centile | 35 ± 3 weeks | SGA $n = 59$ <br> FGR $n = 150$ <br> AG $n = 150$ | Both SGA and FGR had larger hearts, decreased LV sphericity index, decreased TAPSE and increased left MPI |

(*Continued*)

**Table 1. (Continued)**

| Author (year) | Definition of FGR and/or SGA | Gestational age at study | Number of fetuses | Echocardiographic outcomes |
|---|---|---|---|---|
| Rodríguez-López et al. (2017) | SGA: EFW <10th centile divided into three phenotypes: globular, elongated and hypertrophic | ∼30–37 weeks | SGA $n = 126$ (Globular $n = 68$; elongated $n = 37$; hypertrophic $n = 21$) AG $n = 64$ | Elongated and globular SGA phenotypes were associated with late-onset FGR, while hypertrophic phenotype was associated with early-onset FGR and poorest perinatal outcomes. Globular SGA had lowest LV sphericity index. Hypertrophic SGA had highest LV wall thickness. |

AG: appropriately grown; EFW: estimated fetal weight; FGR: fetal growth restriction; GA: gestational age; IVRT: isovolumetric relaxation time; LV: left ventricle; MAPSE: mitral annular plane systolic excursion; MPI: myocardial performance index; PI: pulsatility index; RV: right ventricle; SGA: small for gestational age; TAPSE: tricuspid annular plane systolic excursion; UA: umbilical artery; UtA: uterine artery.

with normal monophasic shortening seen in AG fetuses (Crispi et al., 2014). Post-systolic shortening in the basal segment of the septal ventricular wall was observed in 57% of the SGA cases and in none of the controls (Crispi et al., 2014). Interestingly, a similar pattern was observed post-natally in 2–5-day-old SGA infants, with basal segments being affected more severely (Sehgal et al., 2013). The timing of onset of FGR and/or SGA may also impact the severity of cardiac dysfunction, with a prospective case-control study demonstrating that an earlier SGA diagnosis was associated with greater systolic and diastolic dysfunction during fetal life (Oluklu et al., 2023). These studies highlight significant cardiovascular impairments in FGR and/or SGA fetuses that are established during fetal life, including impaired systolic and diastolic function and altered myocardial contractility (Comas et al., 2010; Crispi et al., 2014; Domínguez-Gallardo et al., 2023; DeVore et al., 2021; Oluklu et al., 2023; Pérez-Cruz et al., 2015; Rodríguez-López et al., 2017). Because of these deficits, much research has also focused on how these changes might impact the cardiopulmonary transition to extrauterine life, and whether they contribute to poor cardiovascular outcomes in the postnatal period.

## Cardiovascular impairments during the transition at birth

The cardiopulmonary transition at birth involves major cardiovascular changes, driven by lung aeration, to enable the lungs to replace the placenta as the main source of gas exchange in the neonate (Hooper et al., 2015). The normal cardiopulmonary transition at birth has been well reviewed (Chakkarapani et al., 2024; Hooper et al., 2015). However, the impact of FGR or SGA on this transition has not been fully elucidated. Maladaptive cardiac remodelling during fetal life may impair the neonate's ability to coordinate the critical increases in biventricular output, closure of the fetal shunts, and the rise in systemic vascular resistance. An inability to achieve these normal physiological adaptations may increase the risk of circulatory failure during early transition, and potentially increase the requirement for circulatory support in the neonatal period (Sehgal et al., 2021).

**Clinical studies on cardiovascular transition at birth in growth-restricted neonates.** At birth, clamping of the umbilical cord removes the low-resistance placental circulation and triggers lung liquid clearance to enable immediate gas exchange within the lung (Hooper et al., 2015). Rapid clearance of lung liquid and aeration of the lung triggers a decrease in pulmonary vascular resistance (PVR), resulting in an increase in pulmonary blood flow (Hooper et al., 2015; Murphy, 2005). As umbilical venous return is abolished with removal of the placenta,

pulmonary blood flow solely contributes to venous return to the left heart, and becomes the primary source of LV preload (Hooper et al., 2015). Immediate cord clamping initially results in a reduction in biventricular output due to a sudden loss in venous return and increase in systemic vascular resistance (SVR) following the loss of the low-resistance placental circulation (Hew & Keller, 2003). Lung aeration restores the loss in LV preload through an increase in pulmonary blood flow, thereby stabilising CO (Hooper et al., 2015). Left ventricular output (LVO) normally increases twofold in the first hour of life to accommodate the increasing oxygenation demands of the systemic circulation, and occurs in tandem with changes within the heart such as increases in SV and fractional shortening (Agata et al., 1991).

Several studies show that growth-restricted infants demonstrate systolic dysfunction in the immediate neonatal period, particularly showing an impaired ability to increase LVO and SV (Table 2). A cohort study of extremely preterm infants by Leïpala et al. found that AG infants increased their LVO between postnatal day 1 and day 14, whereas SGA infants exhibited elevated LVO on day 1 but failed to show further increases thereafter (Leipälä et al., 2003). Notably, approximately a quarter of infants in both cohorts had a significant patent ductus arteriosus, which may have influenced left ventricular dimensions and contributed to altered loading conditions (Leipälä et al., 2003). Similarly, Fouzas et al. reported that preterm AG infants showed an increase in LV SV between postnatal days 2 and 5 (Fouzas et al., 2014). This, along with a decrease in MPI and heart rate, suggests an overall improvement in global myocardial function in AG infants over time (Fouzas et al., 2014). Although FGR infants had a higher LVO on day 1, they demonstrated a decrease in LVO from day 2 to day 5, with no change in LV SV, MPI and HR, thus indicating progressive systolic dysfunction in the immediate postnatal period (Fouzas et al., 2014). However, a study by Ishii et al. contradicts these findings of an initially higher LVO, instead reporting SGA infants to have consistently lower LVO and ejection fraction from 3—6 h to 72 h after birth (Ishii et al., 2014). Another study by Montaldo et al. described lower LVO in FGR infants at 6 and 24 h, followed by similar LVO in both groups at 48 and 72 h (Montaldo et al., 2022). However, impairments in LV function may persist beyond this initial transitional period, where SGA infants demonstrated a persistently higher MPI beyond 48 h of life, indicative of global LV dysfunction (Verma et al., 2023). The variability in findings across these studies is likely due to the heterogeneity in gestational ages among study populations, ranging from <28 weeks to ≥39 weeks, representing a period in which substantial cardiovascular maturation occurs (Jonker et al., 2007, 2010, 2015). This variation in developmental stage may also interact with the timing, duration and severity of FGR, which are all

known to influence cardiovascular function at the time of assessment (Botting et al., 2014; Bubb et al., 2007; Dimasi et al., 2021; Dimasi, Darby, Cho et al., 2023; Louey et al., 2007; Morrison et al., 2007; Thompson et al., 2000; Wang et al., 2011).

In addition to the systolic dysfunction described above, diastolic dysfunction is also evident in FGR and/or SGA neonates. SGA infants assessed during the second week of life demonstrated significantly higher transmitral E/A ratios, isovolumetric relaxation time and ratio of passive transmitral flow to passive myocardial relaxation velocity (E/E' ratio), demonstrating diastolic dysfunction due to impaired myocardial compliance (Sehgal et al., 2017). Impairments in diastolic function can lead to elevated end-diastolic left atrial and pulmonary venous pressures and impede the normal fall in PVR after birth, which was reflected in SGA infants requiring significantly longer durations of respiratory support compared to AG infants (Sehgal et al., 2017). Overall, these data show that growth-restricted neonates present with both systolic and diastolic dysfunction following the transition to extrauterine life, which may increase the risk of inadequate oxygenation and perfusion of systemic and pulmonary circulations during the transitional period.

After birth, a reduction in PVR is critical to establish newborn circulation and to enable the low-resistance pulmonary circulation to accommodate the increase in right ventricular output (RVO) (Hooper et al., 2015). However, growth-restricted infants may have an impeded ability to meet these normal decreases in PVR. A prospective study of preterm SGA neonates 10.5 days after birth showed that SGA infants had a higher baseline PVR compared to AG infants (Sehgal, Gwini et al., 2019). Further, preterm SGA neonates requiring surfactant on the first day of life showed a subdued improvement in PVR compared to AG infants, suggesting impaired vasodilatation and vasoreactivity resulting in abnormal baseline pulmonary vascular function (Sehgal et al., 2020). Similarly, a prospective observational study by Suciu et al. described SGA infants as having a lower pulmonary acceleration time in the first 2 days of life, which is reflective of a failure to normally reduce PVR postnatally (Suciu et al., 2023). This increased PVR occurred in conjunction with a lower TAPSE, indicating impaired RV contractility, demonstrating how persistently increased afterload from elevated PVR impairs RV systolic performance after birth (Suciu et al., 2023). However, a prospective observational study by Kumar et al. suggests that in some cases, the removal of chronically elevated afterload after birth may progressively improve RV dysfunction in the postnatal period (Kumar et al., 2019). In this study, SGA infants had a significantly higher RV MPI on days 1 and 2 of life, suggesting global RV dysfunction; however, this was no longer evident by day 3 (Kumar et al., 2019). However, improvements in MPI

**Table 2. Studies of FGR and/or SGA cardiovascular structure and function during neonatal period**

| Author (year) | Definition of FGR and/or SGA | Postnatal age at study | Number of infants | Cardiovascular outcomes |
|---|---|---|---|---|
| Cohen et al. (2017) | FGR: EFW <10th centile and/or reduced growth velocity + abnormal UA/UtA/MCA Doppler. Preterm infants born <36 weeks GA; term infants born >37 weeks GA | Postnatal day 1; follow-up at 1 month and 6 months | Preterm FGR $n = 25$ AG $n = 41$ (Preterm AG $n = 22$; term AG $n = 19$) | FGR infants had increased heart rate and compromised heart rate variability on postnatal day 1, with reduced low frequency and total power during quiet and active sleep compared to term AG. |
| Cohen et al. (2018) | FGR: EFW <10th centile and/or reduced growth velocity + abnormal multivessel integrated Doppler. Preterm infants born <36 weeks GA; term infants born >37 weeks GA | Postnatal day 1; follow-up at 1 month and 6 months | Preterm FGR $n = 24$ AG $n = 42$ (Preterm AG $n = 23$; term AG $n = 19$) | FGR infants had decreased sphericity index compared to preterm AG on postnatal day 1; no other differences were observed at this time point. |
| Fouzas et al. (2014) | FGR: BW <10th centile + abnormal UA Doppler | Postnatal days 2 and 5 | FGR $n = 30$ AG $n = 30$ | FGR infants had increased interventricular septum hypertrophy and LV dilatation compared to AG. From postnatal day 2 to 5, AG infants experienced a decrease in MPI and HR and an increase in LV SV and LVO; FGR infants experienced a decrease in LVO and no changes in MPI, HR and LV SV. |
| Ishii et al. (2014) | SGA: BW <10th centile and head circumference ±2 SD | 3–6 h, 12 h, 24 h, 48 h and 72 h after birth | SGA $n = 30$ AG $n = 57$ | SGA infants had reduced ejection fraction and LVO at 3–6 h, 24 h, 48 h and 72 h compared to AG infants. |
| Kumar et al. (2019) | SGA: BW <2 SD; infants born between 34–41 weeks | Postnatal days 1 (0–8 h), 2 (24–36 h), and 3 (48–72 h) | SGA $n = 35$ AG $n = 35$ | SGA infants had increased LV MPI, heart rate, LVIDD and LVIDS at all three time points compared to AG. SGA had increased RV MPI on days 1 and 2, but not 3. No differences in fractional shortening, ejection fraction or area shortening were observed. |
| Leipälä et al. (2003) | SGA: BW <1500 g and <2 SD | Postnatal days 1, 2, 3, 4–6 and 7–14. | SGA $n = 31$ AG $n = 31$ | SGA infants had higher LVO than AG infants on postnatal day 1, with no further increases over time. AG infants significantly increased LVO from postnatal days 1 to 7–14. |
| Montaldo et al. (2022) | FGR: EFW <3rd centile or <10th centile + abnormal UA, UtA, MCA Doppler and/or cerebroplacental ratio | 6 h, 24 h, 48 h and 72 h after birth | FGR $n = 105$ AG $n = 105$ | FGR infants had higher SVC flow and lower LVO at 6 h and 24 h. No differences in RVCO were observed at any time point. |
| Patey et al. (2019) | FGR: EFW <10th centile with abnormal Doppler | Fetal scan: 38 weeks GA Neonatal scan: ~0.5 d after birth | FGR $n = 33$ AG $n = 54$ | FGR fetuses had a higher LV sphericity index and higher LV MPI that persisted after birth. FGR neonates showed improvements in RV sphericity index and RV MPI after birth. |

*(Continued)*

**Table 2. (Continued)**

| Author (year) | Definition of FGR and/or SGA | Postnatal age at study | Number of infants | Cardiovascular outcomes |
|---|---|---|---|---|
| Rodriguez-Guerineau et al. (2018) | SGA: BW <10th centile | Postnatal days 3–4 | SGA n = 25 AG n = 25 | SGA infants had lower LV and RV sphericity index, lower TAPSE and higher LV SV compared to AG infants. SGA infants also exhibited lower septal e′ and higher RV MPI. |
| Sehgal et al. (2013) | SGA: BW <3rd centile | Postnatal days 2–5 | SGA n = 20 AG n = 20 | SGA infants had higher blood pressure, lower LVO, and exhibited signs of diastolic dysfunction (reduced E and A velocities, increased E/A ratio, prolonged IVRT). Vascular dysfunction was observed in SGA (increased aortic intima-media thickness, arterial wall stiffness index and input impedence). |
| Sehgal et al. (2017) | SGA: BW <10th centile | Postnatal week 2 | SGA n = 20 AG n = 20 | SGA infants had decreased sphericity index and increased relative wall thicknesses compared to AG. SGA infants exhibited systolic (lower ejection fraction) and diastolic (higher transmitral E/A and IVRT) dysfunction. |
| Sehgal et al. (2018) | SGA: BW <10th centile | Postnatal day 10 | SGA n = 20 AG n = 20 | SGA infants had higher blood pressure, maximum aorta intima-media thickness, arterial wall stiffness and peripheral resistance. |
| Sehgal, Gwini et al. (2019) | SGA: BW <10th centile | Postnatal day 10.5 | SGA n = 20 AG n = 20 | SGA infants had higher PVR, reduced pulmonary artery pulsatility, thicker pulmonary artery inferior wall, lower TAPSE and higher RV MPI |
| Sehgal et al. (2020) | SGA: BW <10th centile | Postnatal day 1 | SGA n = 10 AG n = 20 | SGA infants had higher baseline PVR compared to AG infants. SGA infants had a blunted improvement in PVR, FAC and TAPSE following surfactant administration. |
| Skilton et al. (2005) | SGA: BW <10th centile | Postnatal days 0–4 | SGA n = 25 AG n = 25 | SGA infants had higher maximum aorta intima-media thickness compared to AG. |
| Suciu et al. (2023) | SGA: EFW <10th centile | 24 h and 48 h after birth | SGA n = 18 AG n = 18 | SGA infants had lower TAPSE, lower pulmonary acceleration time, higher PVR and higher LV ejection fraction compared to AG infants at both time points. |
| Verma et al. (2023) | SGA: BW <10th centile | 24–48 h and ≥48 h after birth | SGA n = 24 AG n = 30 | SGA infants had lower fractional shortening, ejection fraction and RVO at first, but not second time point. TAPSE and LV MPI were significantly lower in SGA infants at both time points. |

AG: appropriately grown; BW: birth weight; e′: early diastolic annular peak velocity; EFW: estimated fetal weight; FAC: fractional area change; FGR: fetal growth restriction; GA: gestational age; IVRT: isovolumetric relaxation time; LV: left ventricle; LVO: left ventricular output; LVIDD: left ventricular internal diameter index during diastole; LVIDS: left ventricular internal diameter index during systole; MAPSE: mitral annular plane systolic excursion; MCA: middle cerebral artery; MPI: myocardial performance index; PI: pulsatility index; PVR: pulmonary vascular resistance; RV: right ventricle; SD: standard deviation; SGA: small for gestational age; SV: stroke volume; SVC: superior vena cava; TAPSE: tricuspid annular plane systolic excursion; UA: umbilical artery; UtA: uterine artery.

may not always correlate with improved RVO. A study by Verma et al. described a persistent reduction in RVO and TAPSE in SGA infants beyond the first 2 days of life, suggesting that other factors beyond intrinsic myocardial performance may contribute to impairments in ventricular output (Verma et al., 2023). Collectively, these data show that RV dysfunction is evident in FGR and/or SGA neonates after birth, therefore increasing the risk of inadequate pulmonary perfusion, pulmonary hypertension, and respiratory morbidity in the perinatal period.

Morphological adaptations established during fetal life, such as increased cardiac globularity, may persist beyond birth and into the early postnatal period. A cohort study described greater cardiac sphericity in FGR infants on the first day of life, which persisted until 1 month of age but normalised by 6 months of age (Cohen et al., 2018). Similarly, a prospective cohort study described SGA infants to have more globular hearts at 3–4 days of life, which was associated with a lower TAPSE and lower septal early diastolic annular peak velocity ($e'$), suggesting impaired RV systolic and interventricular septal diastolic function (Rodriguez-Guerineau et al., 2018). Interestingly, a cohort study by Patey et al. suggests that some indices of fetal cardiac dysfunction may not persist in FGR infants after birth (Patey et al., 2019). In this study, indices of cardiac dysfunction detected ∼7.5 days before birth, including a decreased RV sphericity index and increased RV MPI, were improved in the first few hours after birth (Patey et al., 2019). However, persistent LV dysfunction, such as increased LV longitudinal strain and reduced LV torsion, remained evident after birth (Patey et al., 2019). The resolution of RV dysfunction may reflect the removal of the chronically elevated placental vascular resistance after birth, whereas persistent LV dysfunction likely reflects the increasing systemic workload against the left heart after birth.

In addition to echocardiographic indices, FGR newborns are also prone to systemic vascular changes and unstable cardiovascular control during the first few days of life (Table 2). Suciu et al. described higher mean and systolic blood pressures in SGA infants in the first 48 h of life (Suciu et al., 2023). Similar results were described by Sehgal et al., with SGA infants demonstrating higher systolic, diastolic and mean blood pressures in the first 3 days of life (Sehgal et al., 2013). Blood pressure changes are likely indicative of vascular adaptations, as seen in an observational study of SGA infants in the second week of life, demonstrating higher systolic blood pressure accompanied by increased peripheral artery resistance, wall stiffness and impedance (Sehgal et al., 2018). Changes to the composition of the aorta were described in a cross-sectional study of 0- to 4-day-old SGA infants, where an increase in aortic intima-media thickness highlights an increased risk of atherosclerosis later in life (Skilton et al., 2005). Haemodynamic instability has also

been described in preterm FGR infants on the first day of life, where higher heart rates and reduced heart rate variability indicated compromised autonomic cardiovascular control (Cohen et al., 2017). Haemodynamic instability increases the risk of cerebrovascular injury, particularly in preterm infants with a reduced capacity for cerebral autoregulation (du Plessis, 2009). Taken together, these data suggest that instabilities in blood pressure and cardiovascular control, although subclinical, may reflect early vascular maladaptation in growth-restricted neonates and potentially contribute to an increased risk of hypertension later in life. Overall, the inability to adequately increase stroke volume and cardiac output, combined with persistent cardiac globularity and unstable blood pressures, suggests that the growth-restricted neonate has an impaired ability to tolerate the normal haemodynamic adaptations at birth. These cardiac impairments may increase the risk of cardiovascular, respiratory and neurological morbidities in the neonatal period, and may render the neonate more vulnerable to additional antenatal stressors present during this period. However, clinical data can only provide speculative information on the cardiovascular transition of compromised infants, highlighting the need for preclinical models that allow direct investigation of the underlying physiological mechanisms during an impaired cardiovascular transition.

**Preclinical studies.** There is a paucity of preclinical data regarding the effects of the cardiopulmonary transition at birth on the growth-restricted heart, with most studies reporting cardiovascular dysfunction either before birth or at much older postnatal ages equivalent to adolescent or adult cardiac development, with the current gap in knowledge summarised in Fig. 2.

Despite differences in timing, severity and method of FGR induction, many preclinical studies consistently reproduce the *in utero* cardiovascular dysfunction observed in human FGR fetuses (Fig. 2). These include echocardiographic studies in rodents demonstrating fetal systolic and diastolic dysfunction (Dai et al., 2021; Thompson et al., 2020) that persist into adolescence and adulthood (Alsaied et al., 2017; Coats et al., 2021; Kumar et al., 2020; Rueda-Clausen et al., 2009; Shah et al., 2017; Thompson et al., 2018). Three preclinical studies have investigated echocardiographic function immediately after birth in FGR models. One study assessed lambs delivered at 0.85 gestation, where FGR was induced by single umbilical artery ligation (SUAL) at 0.7 gestation, resulting in placental insufficiency, chronic hypoxaemia, and asymmetrical growth restriction (i.e. brain sparing) (Polglase et al., 2016; Sutherland et al., 2024). These lambs showed lower LVO, reduced cerebral blood flow and higher systemic vascular resistance after immediate

cord clamping, suggesting an impaired initial cardio-pulmonary transition at birth (Polglase et al., 2016). However, no echocardiographic studies have investigated fetal or adolescent/adult time points in sheep. In FGR rat pups induced by chronic maternal hypoxia (11.5% $O_2$) from 0.71 gestation until term, increased aortic stiffness and decreased heart rate were observed on the first day of life despite echocardiographic parameters being comparable to AG offspring (Kumar et al., 2020). Similarly, near-term FGR rat pups induced by unilateral uterine artery ligation at 0.8 gestation exhibited significant LV dysfunction evident 2–3 days after birth, including increased LV wall thickness, reduced MPI and reduced fractional shortening (Dai et al., 2021).

Echocardiographic and *in vivo* studies are informative but limited by autonomic, hormonal, and circulatory influence on heart function. In contrast, *ex vivo* heart models enable assessment of FGR-induced cardiac dysfunction independently of preload, afterload and systemic regulation (Fletcher et al., 2005). Studies conducted during the fetal period (Tare et al., 2012, 2014) and adolescent/adult stages (Giussani et al., 2012; Hansell et al., 2022) have reported contradicting information regarding the effect of FGR on *ex vivo* function, particularly left ventricular contractility. Regardless, no *ex vivo* studies have investigated the impact of FGR immediately after the cardiopulmonary transition at birth.

FGR has been shown to disrupt normal fetal cardio-myocyte development across multiple species, including sheep (Bubb et al., 2007; Louey et al., 2007; Jonker et al., 2018; Morrison et al., 2007; Zhang et al., 2024) and rodents (Corstius et al., 2005). Most studies report reduced cell cycle activity, decreased binucleation, and decreased total cardiomyocyte endowment, which is particularly significant as cardiomyocyte endowment is largely fixed at birth and limits the heart's capacity for postnatal growth and repair (Bergmann et al., 2015; Olivetti et al.,

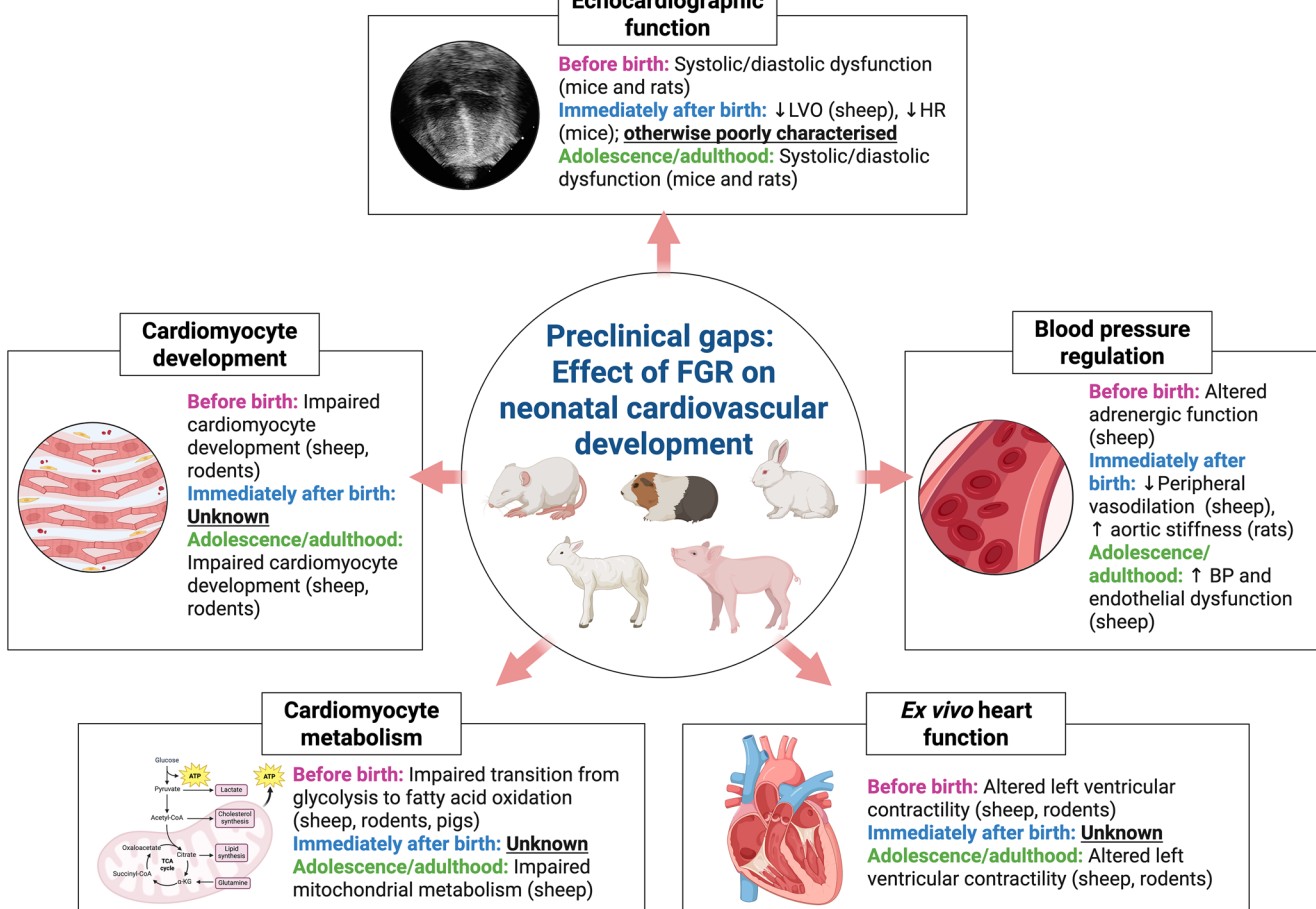

**Figure 2. Preclinical gaps on the effect of FGR on neonatal cardiovascular dysfunction**
Preclinical FGR studies in rodents, sheep and pigs have identified alterations in echocardiographic function, cardio-myocyte development, blood pressure regulation, cardiomyocyte metabolism and *ex vivo* heart function before birth and later in postnatal life. However, the immediate postnatal period remains poorly characterised, thus limiting the current understanding of the impact of FGR on the cardiovascular transition at birth.

1991). These deficits persist in sheep and are evident 21 days (Wang et al., 2011) and 1 year after birth (Vranas et al., 2017), which are equivalent to a 6- to 9-month-old infant and an adult human, respectively. Similarly, FGR guinea pigs raised into adulthood also demonstrate impaired cardiomyocyte development (Botting et al., 2018; Masoumy et al., 2018). While data on the immediate postnatal period are limited, current evidence supports the view that early deficits in cardiomyocyte endowment continue to affect long-term cardiac development.

Vascular dysfunction is increasingly recognised as a key feature of FGR and is demonstrated across the lifespan in several species. In rodents, FGR is associated with increased vascular stiffness and contractility of the femoral artery before birth (Cañas et al., 2017), with lasting impairments in femoral artery vasodilatation and heightened vasoconstrictor responses observed in adulthood (Giussani et al., 2012; Graton et al., 2024; Krause et al., 2019; Thompson et al., 2014). Aortic stiffness has also been observed in FGR sheep before birth (Thompson, Gros et al., 2011; Thompson, Richardson et al. 2011). Interestingly, coronary arteries from FGR fetuses induced by placental embolization from 0.74–0.88 gestation, resulting in chronic hypoxaemia, exhibited heightened vasoconstrictor responses but retained increased passive arterial compliance (Bubb et al., 2007). This combination of functional and mechanical adaptations therefore reflects a compensatory mechanism to preserve myocardial perfusion under hypoxic conditions (Bubb et al., 2007). Meanwhile, FGR lambs raised into adulthood show enhanced femoral and mesenteric artery vasoconstriction and reduced endothelium-dependent and endothelium-independent vasodilatation (Allison, Kaandorp et al., 2016; Brain et al., 2019; Botting et al., 2020; Hansell et al., 2022). While these studies collectively support the presence of long-term FGR-mediated vascular dysfunction, the immediate postnatal period remains largely understudied, with the exception of three sheep studies. Coronary arteries from term 24-hour-old FGR lambs induced by SUAL at 0.72 gestation showed impaired endothelial function and increased vascular tone characterised by increased passive wall stiffness, heightened vasoconstriction in response to the thromboxane mimetic U46619, and reduced endothelium-dependent vasodilatation due to impaired sensitivity to bradykinin and nitric oxide (NO) bioavailability (Tare et al., 2014). The aforementioned study by Polglase *et al.* demonstrated reduced sensitivity to sodium nitroprusside and heightened responsiveness to methacholine in the femoral arteries of FGR lambs, indicating impaired vasodilatation (Polglase et al., 2016). Early vascular dysfunction was also observed in newborn FGR lambs delivered at 0.92 gestation following SUAL at 0.6 gestation (Rock et al., 2023). In these lambs, progressive impairments in femoral artery endothelium-dependent vasodilatation were observed at 24 hours of age and persisted until 4 weeks of age, suggesting that early vascular dysfunction may worsen postnatally (Rock et al., 2023).

Cardiomyocytes must also switch from a glycolytic to fatty acid oxidative state around the time of birth, which is a process driven by mitochondrial maturation to support increasing energy demands and changing substrate availability and oxygen requirements after birth (reviewed in depth by Dimasi, Darby & Morrison et al., 2023). However, FGR results in impaired cardiac metabolism before birth in sheep (Chang et al., 2024; Dimasi et al., 2021; Dimasi, Darby, Cho et al., 2023; Drake et al., 2022), rabbits (Gonzalez-Tendero et al., 2013; Simões et al., 2018), and rodents (Maréchal et al., 2021; Song et al., 2021; Thompson et al., 2018). Cardiac metabolism remains impaired into adulthood in rodents (Rueda-Clausen, Morton et al., 2011; Thompson et al., 2018, 2019). Evidence from 21-day-old FGR lambs shows increased glucose reliance and altered IGF-2/IGF-2R signalling, contributing to inefficient energy metabolism and early markers of pathological cardiac hypertrophy (Wang et al., 2011, 2013, 2015). Despite these findings, it remains largely unclear whether newborn FGR hearts have successfully completed this critical metabolic maturation immediately after the cardiopulmonary transition at birth.

Overall, these clinical and preclinical data suggest that morphological and functional adaptations in the heart and peripheral vasculature underlie potential impairments of the growth-restricted cardiovascular system to adapt from fetal to neonatal life. Despite this, more preclinical research is still needed to bridge the current knowledge gap regarding the mechanisms and potential treatments underlying these impairments in FGR infants. This is important as several clinical studies have suggested that these impairments may persist beyond the transitional period, with cardiac dysfunction still evident in the first few months of life (Altın et al., 2012; Änghagen et al., 2022; Cruz-Lemini et al., 2016; Czernik et al., 2013; Gürses & Seyhan, 2013). Additionally, the variability in echocardiographic and vascular findings highlights the need for a consistent indicator of cardiovascular dysfunction, such as biomarkers, to guide the clinical management of growth-restricted infants during the transitional period.

## Biomarkers of cardiac damage at birth

There is growing evidence to suggest that biochemical markers of cardiac damage can be detected in the cord blood of FGR and/or SGA infants around the time of birth, suggesting underlying myocardial stress that may impair the transition at birth. One such biomarker is

B-type natriuretic peptide (BNP), which is a hormone secreted by ventricular cells in response to increased ventricular pressure load and wall stress (Alter et al., 2007). Several studies have shown increased BNP in the cord blood of growth-restricted infants at delivery, with BNP levels increasing in proportion to the severity of FGR (Crispi et al., 2008; Perez-Cruz et al., 2018). However, this increase in BNP was not observed in SGA infants compared to AG infants (Perez-Cruz et al., 2018). The inactive N-terminal fragment of BNP, NT-proBNP, is also a known strong predictor of cardiovascular dysfunction in adulthood (Rudolf et al., 2020), and is elevated in the cord blood of FGR infants proportional to severity (Girsen et al., 2007). NT-proANP, another marker of cardiac dysfunction, is increased in the cord blood of FGR infants, and is positively correlated with worsening umbilical artery Doppler (Mäkikallio et al., 2002). Cardiac troponin is a regulator of $Ca^{2+}$-mediated interactions between actin and myosin, but is also released by the heart as an indicator of cardiomyocyte compromise (Vijlbrief et al., 2012). Several studies have demonstrated increased levels of two subunits of the troponin complex, cardiac troponin I and T, in the cord blood of FGR infants (Mäkikallio et al., 2002; Perez-Cruz et al., 2018; Yakıştıran et al., 2019). Importantly, SGA infants also demonstrated an increase in troponin I compared to AG infants, suggesting that subclinical myocardial injury may still be present in these infants (Chaiworapongsa et al., 2002; Perez-Cruz et al., 2018). However, a study by Crispi *et al.* showed no differences in cord blood troponin I levels between FGR and AG infants despite FGR infants having altered echocardiographic function (Crispi et al., 2008). This variability across studies may be explained by the reduced sensitivity of cardiac troponins as a marker of myocardial damage in neonates, despite being highly sensitive in adults (Karlén et al., 2019). FGR cord blood has also been shown to express higher levels of heart-type fatty acid-binding protein (H-FABP), which is involved in fatty acid metabolism and is a known marker of acute myocardial cell damage (Crispi et al., 2008). These data show the potential value of combining biomarkers with echocardiographic indices to determine infants at greater risk of cardiovascular compromise during the perinatal period.

## Risk to the heart with a secondary insult at birth

As growth-restricted infants exhibit impaired cardiovascular transition and increased biomarkers of cardiac injury at birth, they may be more susceptible to additional stressors during the early postnatal period. These stressors may include exposure to secondary hypoxic events, such as umbilical cord occlusion or birth asphyxia, or cardiovascular instability causing fluctuations in blood pressure. Regardless of the aetiology, preexisting cardiac adaptations to chronic hypoxia may render the growth-restricted neonate more vulnerable to secondary insults. This concept aligns with the 'double hit' hypothesis, where exposure to *in utero* chronic hypoxia programmes the fetus with latent vulnerabilities that may not manifest until a secondary challenge occurs. Preclinical studies have shown that FGR offspring are more susceptible to adverse cardiopulmonary outcomes with ageing, specifically increased myocardial injury following ischaemia-reperfusion (Rueda-Clausen et al., 2009) and metabolic dysfunction in response to a high-fat diet (Rueda-Clausen, Dolinsky et al., 2011). These programmed susceptibilities are mediated by disrupted signalling pathways, such as PKC$\varepsilon$ repression and impaired insulin signalling, and may occur in a sex-specific manner (Rueda-Clausen, Dolinsky et al., 2011; Xue & Zhang, 2009). Additionally, clinical cohort studies have described not only a greater prevalence of birth asphyxia in growth-restricted infants (Groene et al., 2021; Liu et al., 2014; Wood et al., 2021), but also that asphyxiated FGR infants present with worse outcomes than AG infants after asphyxia, such as moderate to severe metabolic acidosis (Low et al., 1972). Together, these findings highlight the vulnerability of the FGR infant to additional hits, which may increase their propensity for further cardiovascular dysfunction.

Much of our current understanding of the ability of the FGR neonate to respond to secondary physiological insults, such as birth asphyxia or cardiovascular instability, is derived from preclinical studies investigating *in vivo* haemodynamic responses in sheep. The secondary insults from these studies have largely focused on exposure to maternal hypoxia or direct manipulation of blood pressure as a secondary insult to the fetus. In one such model, FGR fetal sheep exposed to placental embolization from ∼0.8 gestation demonstrated a heightened circulatory defence response to brief maternal isocapnic hypoxia (10% $O_2$) at 0.89–0.95 gestation (Block et al., 1984). This response was characterised by pronounced perfusion to vital organs such as the brain, heart and adrenal glands compared to control fetuses (Block et al., 1984). However, when the period of placental embolization was reduced from 2 weeks to 9 days, acute hypoxia did not result in differences in the circulatory response between the FGR and control groups, suggesting that the degree of severity or duration of FGR may greatly influence the ability of the neonate to adequately adapt to a secondary insult (Block et al., 1989). Similarly, FGR fetal sheep induced by placental carunclectomy, a model involving the removal of majority of the placental caruncles prior to conception resulting in chronic hypoxaemia and hypoglycaemia, were investigated at 0.91–0.95 gestation (Robinson et al., 1983). When exposed to acute maternal isocapnic hypoxia (9% $O_2$), FGR fetuses displayed only

modest physiological changes, including increased hypoxaemia and acidaemia, and a faster heart rate recovery period compared to control fetuses (Robinson et al., 1983). Additionally, FGR fetuses induced by placental carunclectomy showed increased ductus venosus and foramen ovale flow during maternal hypoxia (reduction in maternal $S_{pO_2}$ to 80%–85%), indicating preferential shunting of oxygen-rich blood to maintain cerebral oxygen delivery (Darby et al., 2024). Exposure to high altitudes during pregnancy, resulting in hypobaric hypoxia, is also known to cause FGR (Brown & Giussani, 2024). In contrast to other models of FGR, high-altitude fetuses (3600 m above sea level for the entirety of gestation) showed a blunted cardiovascular response to secondary maternal hypoxia (10% $O_2$ for 1 h) at 0.9 gestation, with an impaired ability to alter umbilical and carotid blood flow compared to sea-level control fetuses. Interestingly, even in the absence of growth restriction, spontaneous fetal compromise (chronic hypoxaemia, hypoglycaemia or acidaemia) results in enhanced vasoconstrictor responses during maternal isocapnic hypoxia (9% $O_2$ for 1 h) at 0.79–0.84 gestation (Gardner et al., 2002). However, only chronically hypoxaemic fetuses exhibited an increase in noradrenaline release and greater chemoreflex sensitivity during this period (Gardner et al., 2002).

Several studies also investigate secondary insults beyond acute hypoxia, including those that induce cardiovascular instability by direct manipulation of blood pressure. For example, FGR fetal sheep (induced by maternal exposure to 10% $O_2$ from 0.81 gestation) were exposed to an acute hypotensive challenge through intravenous administration of the vasodilator sodium nitroprusside at 0.88 gestation (Allison et al., 2020). In this study, FGR fetuses demonstrated an impaired ability to restore blood pressure homeostasis due to an impaired cardiac baroreflex function with absent sympathetic regulation and a diminished peripheral vasoconstrictor response to the $\alpha_1$-adrenergic agonist phenylephrine (Allison et al., 2020). However, carunclectomy-induced FGR fetuses had a greater dependence on $\alpha$-adrenergic activation to maintain blood pressure in response to $\alpha$-adrenergic receptor blockade at 0.83–0.85 gestation (Danielson et al., 2005). Additionally, this increased reliance on $\alpha$-adrenergic activation was not due to increased post-ganglionic sympathetic activation (Darby et al., 2021). These differences in adrenergic function are likely attributable to the specific method of FGR induction, with maternal hypoxia potentially causing chronic receptor desensitisation, while placental carunclectomy may preserve more stable receptor function. Although the FGR fetus maintains mean arterial pressure through peripheral vasoconstriction, inhibition of NO synthesis at ~0.9 gestation did not elicit an exaggerated hypertensive response in carunclectomy-induced FGR fetuses compared to control fetuses, suggesting that NO does not play an enhanced compensatory role in chronically hypoxaemic fetuses (Dyer et al., 2009). In spontaneously chronically hypoxaemic and growth-restricted fetal sheep, repeated brief umbilical cord occlusions (UCO) initially resulted in faster falls in fetal heart rate, suggesting an earlier onset of cardiovascular instability compared to control fetuses (Lear et al., 2023). Interestingly, these brief repeated UCOs are consistent with contractions during early labour, suggesting that FGR infants may be more vulnerable even during normal birth-related events (Lear et al., 2023). Impairments in the fetal cardiovascular defence to a secondary challenge may also persist postnatally, as evidenced in a study where 0.85 gestation FGR lambs (induced by SUAL at 0.6 gestation) were exposed to a mild asphyxia by complete UCO until mean arterial pressure declined to 25 mmHg (Oyang et al., 2023). In this study, FGR lambs appeared to have a greater tolerance to asphyxia compared to controls, as shown by a longer duration to reach the blood pressure threshold, greater maintenance of carotid blood flow during asphyxia, and fewer resuscitation requirements, such as chest compressions (Oyang et al., 2023). However, FGR lambs had a sustained blood pressure overshoot in the post-asphyxia period, suggesting a potential risk for cerebrovascular damage (Oyang et al., 2023). Taken together, these preclinical studies highlight that growth-restricted fetuses have altered physiological responses to acute secondary challenges, such as acute hypoxia, hypotension and asphyxia. However, there is still a lack of understanding of the mechanisms mediating these altered cardiovascular responses, and whether there is an increased propensity of FGR infants in developing cardiac injury following these secondary insults.

## Conclusion

In summary, the cardiopulmonary transition at birth represents a critical window during which the infant must rapidly adapt from intrauterine to extrauterine life. Evidence of subclinical cardiac dysfunction in FGR and SGA infants suggests that these normal adaptations may be compromised and can be detected by echocardiography or cardiac injury biomarkers around the time of birth. The physiological demands of the transition at birth may also render these neonates vulnerable to secondary insults or comorbidities, further impeding their ability to achieve normal physiological adaptations. Overall, these findings highlight that cardiovascular adaptations during fetal development can impede the normal cardiopulmonary transition at birth, placing the growth-restricted neonate at risk of long-term cardiovascular dysfunction.

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

## Additional information

### Competing interests

The authors have no competing interests to declare.

### Author contributions

Z.A. conceptualised and drafted the manuscript. Z.A., A.S., S.L.M, K.J.B., G.R.P. and B.J.A. edited and critically revised the manuscript for intellectual content. All authors have approved the final version of this manuscript and agree to be accountable for all aspects of the work in ensuring that questions related to the accuracy and integrity of any part of the work are appropriately investigated and resolved. All persons designated as authors qualify for authorship, and all those who qualify for authorship are listed.

### Funding

G.R.P. and B.J.A. are supported by National Health and Medical Research Council of Australia (NHMRC) Investigator Grants (Funding ID 1173 731 and 1 175 843, respectively).

### Acknowledgements

Open access publishing facilitated by Monash University, as part of the Wiley - Monash University agreement via the Council of Australian University Librarians.

### Keywords

birth, cardiovascular, cardiopulmonary transition, fetal growth restriction, small for gestational age

## Supporting information

Additional supporting information can be found online in the Supporting Information section at the end of the HTML view of the article. Supporting information files available:

**Peer Review History**

