## [Peer Review History · The Journal of Physiology]

Compromised Cardiopulmonary Transition in Fetal Growth Restricted and Small for Gestational Age Neonates

Zahrah Azman, Arvind Sehgal, Suzanne L Miller, Kristen J Bubb, Graeme R Polglase, and Beth J Allison
DOI: 10.1113/JP289441

Corresponding author(s): Zahrah Azman (siti.azman@monash.edu)

The following individual(s) involved in review of this submission have agreed to reveal their identity: Mitchell C Lock (Referee #2)

Review Timeline:

Submission Date:	16-Jun-2025
Editorial Decision:	28-Jul-2025
Revision Received:	05-Aug-2025
Accepted:	21-Aug-2025

Senior Editor: Laura Bennet

Reviewing Editor: Janna Morrison

Transaction Report:

Dear Ms Azman,

Re: JP-TR-2025-289441 "**Compromised Cardiopulmonary Transition in Fetal Growth Restricted and Small for Gestational Age Neonates**" by Zahrah Azman, Arvind Sehgal, Kristen J Bubb, Graeme R Polglase, and Beth J Allison

Thank you for submitting your manuscript to The Journal of Physiology. It has been assessed by a Reviewing Editor and by 1 expert referee and we are pleased to tell you that it is acceptable for publication following satisfactory revision.

ABSTRACT FIGURES: Authors may use The Journal's premium BioRender account to create/redraw their Abstract Figures (and any other suitable schematic figure). Information on how to access this account is here: <https://physoc.onlinelibrary.wiley.com/journal/14697793/biorender-access>.

REVISION CHECKLIST: Upload a full Response to Referees file. To create your 'Response to Referees' copy all the reports, including any comments from the Senior and Reviewing Editors, into a Microsoft Word, or similar, file and respond to each point, using font or background colour to distinguish comments and responses and upload as the required file type.

We look forward to receiving your revised submission.

Yours sincerely,

Laura Bennet
Senior Editor

REQUIRED ITEMS

- Please include an Abstract Figure file, as well as the Figure Legend text within the main article file. The Abstract Figure is a piece of artwork designed to give readers an immediate understanding of the Review Article and should summarise the main conclusions. If possible, the image should be easily 'readable' from left to right or top to bottom. It should show the physiological relevance of the Review so readers can assess the importance and content of the article. Abstract Figures should not merely recapitulate other figures in the Review. Please try to keep the diagram as simple as possible and without superfluous information that may distract from the main conclusion of the Review. Abstract Figures must be provided by authors no later than the revised manuscript stage and should be uploaded as a separate file during online submission labelled as File Type 'Abstract Figure'. Please ensure that you include the figure legend in the main article file. All Abstract Figures will be sent to a professional illustrator for redrawing and you may be asked to approve the redrawn figure before your paper is accepted.

- Your MS must include a complete "Additional information section" with the following 4 headings and content:

Competing Interests: A statement regarding competing interests. If there are no competing interests, a statement to this effect must be included. All authors should disclose any conflict of interest in accordance with journal policy.

Author contributions: Each author should take responsibility for a particular section of the study and have contributed to writing the paper. Acquisition of funding, administrative support or the collection of data alone does not justify authorship; these contributions to the study should be listed in the Acknowledgements. Additional information such as 'X and Y have contributed equally to this work' may be added as a footnote on the title page.

It must be stated that all authors approved the final version of the manuscript and that all persons designated as authors qualify for authorship, and all those who qualify for authorship are listed.

Funding: Authors must indicate all sources of funding, including grant numbers. If authors have not received funding, this must be stated.

It is the responsibility of authors funded by RCUK to adhere to their policy regarding funding sources and underlying research material. The policy requires funding information to be included within the acknowledgement section of a paper. Guidance on how to acknowledge funding information is provided by the Research Information Network. The policy also requires all research papers, if applicable, to include a statement on how any underlying research materials, such as data, samples or models, can be accessed. However, the policy does not require that the data must be made open. If there are considered to be good or compelling reasons to protect access to the data, for example commercial confidentiality or legitimate sensitivities around data derived from potentially identifiable human participants, these should be included in the statement.

Acknowledgements: Acknowledgements should be the minimum consistent with courtesy. The wording of acknowledgements of scientific assistance or advice must have been seen and approved by the persons concerned. This section should not include details of funding.

- Please upload separate high quality figure files via the submission form.

- Author profile(s) must be uploaded via the submission form. Authors should submit a short biography (no more than 100 words for one author or 150 words in total for two authors) and a portrait photograph of the two leading authors on the paper. These should be uploaded and clearly labelled together in a Word document with the revised version of the manuscript. Any standard image format for the photograph is acceptable, but the resolution should be at least 300 DPI and preferably more. A group photograph of all authors is also acceptable, providing the biography for the whole group does not exceed 150 words.

- It is the authors' responsibility to obtain any necessary permissions to reproduce previously published material and to list these within the main article file. For information, please see: https://jp.msubmit.net/cgi-bin/main.plex?form_type=display_requirements#permissions.

EDITOR COMMENTS

Reviewing Editor:

This review summarises the impact of FGR on the cardiopulmonary transition at birth. This is an important area of study and thus deserves a comprehensive review with reference to major works in the field over decades.

The authors use FGR/SGA throughout the review suggesting that these terms are synonymous. They are not. SGA is a normal baby that was always meant to be a small baby. A FGR baby is one that should have been bigger but has not reached its genetic potential for growth, depending on the cause for not reaching its genetic growth potential, that baby may have undergone adaptations to survive. One of these that is used clinically to distinguish between FGR and SGA is Doppler. The problem is that it is difficult to distinguish between FGR and SGA, not that they are both pathological. It is very helpful that the table indicates whether the original studies classified the babies as SGA or FGR.

Line 66-70 - Is there original work that can be cited here as well?

Line 70-71 - 'at birth' is used twice.

Line 79 - The term hypoxia is used here; however, the sentence is referring to hypoxemia. When the ewe breaths air that is low in oxygen, she is in a hypoxic environment. When blood oxygen is low, hypoxemia is present.

Line 89 - FGR is associated with brainsparing, but there is not always an increase in blood flow to the brain (Miller et al., 2009; Poudel et al., 2015; Zhu et al., 2016; Darby et al., 2024). However, there may be a distribution of regional blood flow within the brain that supports clinical findings of increased blood flow velocity in the MCA.

Line 97 - Studies have been performed to understand the impact of chronic hypoxia on the brainstem. These should be cited (Adams et al., 2001).

Line 101-102 - References to the impact of FGR on the fetal heart should be used to support this statement. These may be clinical or preclinical findings in a range of species but sheep and guinea pig may be most relevant.

Line 120 - This was shown much earlier than 2016. Please cite original work (Gilbert, 1980; Van Hare et al., 1990; Teitel et al., 1991).

It is generally preferred to use 'that' rather than 'which'. However, when 'which' is used, it should be preceded by a comma except when used as 'by which', 'in which', etc.

Others have previously suggested that timing, duration and severity of the causes of FGR impacts how the heart and other organs adapt to FGR (Morrison, 2008; Darby et al., 2020b).

Line 193 - The sentence that ends on this line could be supported by a range of references.

Line 249 - The use of the term 'varying' is quite vague. It is not clear if this refers to the fact that normal cardiac maturation occurs during this time (Jonker et al., 2006; Jonker et al., 2010; Jonker et al., 2015) or if FGR can impact that maturation depending upon the timing, duration and severity of the cause of the FGR (Thompson et al., 2000a; Thompson et al., 2000b; Thompson & Dong, 2005; Bubb et al., 2007; Louey et al., 2007; Morrison et al., 2007; Oh et al., 2008; Thompson et al., 2008; Thompson et al., 2011d; Wang et al., 2011; Evans et al., 2012a; Evans et al., 2012b; Wang et al., 2012a; Wang et al., 2012b; Wang et al., 2012c; Al-Hasan et al., 2013; Thompson et al., 2013; Wang et al., 2013; Al-Hasan et al., 2014; Botting et al., 2014; Wang et al., 2015a; Wang et al., 2015b; Wang et al., 2017; Botting et al., 2018; Darby et al., 2018; Masoumy et al., 2018; Dimasi et al., 2021; Dimasi et al., 2023; Dimasi et al., 2024a; Dimasi et al., 2024b).

Line 340 - this sentence could be supported with references.

Line 341 - The authors introduce a preclinical model of FGR in sheep. There are several sheep models of FGR. It would be useful to describe this model e.g. is there placental insufficiency? FGR? Chronic hypoxemia? Brainsparing?.

Line 349 - There are many studies of the impact of FGR on aortic stiffness (Thompson et al., 2011a; Thompson et al., 2011b; Thompson et al., 2011c; Thompson et al., 2014).

Line 350 - Please reword this sentence to more clearly indicate that FGR was induced by In rats.

Line 357 - It would be useful to more clearly articulate the gaps in knowledge. This could be summarised in a figure.

Line 359 - ... impairments to persist....

It may well be that biomarkers will play an important role in improving the ability to detect FGR. However, the discussion on the challenges of distinguishing between SGA and FGR and the clinical benefit of this should be outlined in the review prior to this section. The mechanistic underpinning of the discussed biomarkers should also be clear earlier in the piece.

It is true that some fetuses are more vulnerable to a second hit. The concept was outlined some time ago and likely applies to many scenarios, including this one. It would be appropriate to acknowledge this (Rueda-Clausen et al., 2009; Xue & Zhang, 2009; Rueda-Clausen et al., 2011).

Line 402-418 - There are other studies that have investigated the impact of additional insults in FGR and chronic hypoxemia (Gardner et al., 2002; Darby et al., 2020a; Darby et al., 2024).

Line 425 -427 - There are papers that have studied the impact of FGR on regulation of blood pressure in the fetus (Ruijtenbeek et al., 2000; Danielson et al., 2005; Dyer et al., 2009).

References

- Adams MB, Brown RE, Gibson C, Coulter CL & McMillen IC. (2001). Tyrosine hydroxylase protein content in the medulla oblongata of the foetal sheep brain increases in response to acute but not chronic hypoxia. *Neurosci Lett* 316, 63-66.
- Al-Hasan YM, Evans LC, Pinkas GA, Dabkowski ER, Stanley WC & Thompson LP. (2013). Chronic hypoxia impairs cytochrome oxidase activity via oxidative stress in selected fetal Guinea pig organs. *Reprod Sci* 20, 299-307.
- Al-Hasan YM, Pinkas GA & Thompson LP. (2014). Prenatal Hypoxia Reduces Mitochondrial Protein Levels and Cytochrome c Oxidase Activity in Offspring Guinea Pig Hearts. *Reprod Sci* 21, 883-891.
- Botting KJ, Loke XY, Zhang S, Andersen JB, Nyengaard JR & Morrison JL. (2018). IUGR decreases cardiomyocyte endowment and alters cardiac metabolism in a sex and cause of IUGR specific manner. *Am J Physiol Regul Integr Comp Physiol*.
- Botting KJ, McMillen IC, Forbes H, Nyengaard JR & Morrison JL. (2014). Chronic hypoxemia in late gestation decreases cardiomyocyte number but does not change expression of hypoxia-responsive genes. *J Am Heart Assoc* 3, pii: e000531.
- Bubb KJ, Cock ML, Black MJ, Dodic M, Boon WM, Parkington HC, Harding R & Tare M. (2007). Intrauterine growth restriction delays cardiomyocyte maturation and alters coronary artery function in the fetal sheep. *J Physiol* 578, 871-881.
- Danielson L, McMillen IC, Dyer JL & Morrison JL. (2005). Restriction of placental growth results in greater hypotensive response to α -adrenergic blockade in fetal sheep during late gestation. *J Physiol* 563, 611-620.
- Darby JRT, McMillen IC & Morrison JL. (2018). Maternal undernutrition in late gestation increases IGF2 signalling molecules and collagen deposition in the right ventricle of the fetal sheep heart. *J Physiol* 596, 2345-2358.
- Darby JRT, Saini BS, Holman SL, Hammond SJ, Perumal SR, Macgowan CK, Seed M & Morrison JL. (2024). Acute-on-chronic: using magnetic resonance imaging to disentangle the haemodynamic responses to acute and chronic fetal hypoxaemia. *Front Med (Lausanne)* 11, 1340012.
- Darby JRT, Varcoe TJ, Holman SL, McMillen IC & Morrison JL. (2020a). The reliance on α -adrenergic receptor stimuli for blood pressure regulation in the chronically hypoxaemic fetus is not dependent on post-ganglionic activation. *J Physiol* 599, 1307-1318.
- Darby JRT, Varcoe TJ, Orgeig S & Morrison JL. (2020b). Cardiorespiratory consequences of intrauterine growth restriction: Influence of timing, severity and duration of hypoxaemia. *Theriogenology* 150, 84-95.
- Dimasi CG, Darby JRT, Cho SKS, Saini BS, Holman SL, Meakin AS, Wiese MD, Macgowan CK, Seed M & Morrison JL. (2024a). Reduced in utero substrate supply decreases mitochondrial abundance and alters the expression of metabolic signalling molecules in the fetal sheep heart. *J Physiol* 602, 5901-5922.
- Dimasi CG, Darby JRT, Holman SL, Quinn M, Meakin AS, Seed M, Wiese MD & Morrison JL. (2024b). Cardiac growth patterns and metabolism before and after birth in swine: Role of miR in proliferation, hypertrophy and metabolism. *J Mol Cell Cardiol Plus* 9, 100084.
- Dimasi CG, Darby JRT & Morrison JL. (2023). A change of heart: understanding the mechanisms regulating cardiac proliferation and metabolism before and after birth. *J Physiol* 601, 1319-1341.
- Dimasi CG, Lazniewska J, Plush SE, Saini BS, Holman SL, Cho SKS, Wiese MD, Sorvina A, Macgowan CK, Seed M, Brooks DA, Morrison JL & Darby JRT. (2021). Redox ratio in the left ventricle of the growth restricted fetus is positively correlated with cardiac output. *J Biophotonics*, e202100157.
- Dyer JL, McMillen IC, Warnes KE & Morrison JL. (2009). No Evidence for an Enhanced Role of Endothelial Nitric Oxide in the Maintenance of Arterial Blood Pressure in the IUGR Sheep Fetus. *Placenta* 30, 705-710.
- Evans LC, Liu H, Pinkas GA & Thompson LP. (2012a). Chronic hypoxia increases peroxynitrite, MMP9 expression, and collagen accumulation in fetal guinea pig hearts. *Pediatr Res* 71, 25-31.
- Evans LC, Liu H & Thompson LP. (2012b). Differential effect of intrauterine hypoxia on caspase 3 and DNA fragmentation in fetal guinea pig hearts and brains. *Reprod Sci* 19, 298-305.
- Gardner DS, Fletcher AJ, Bloomfield MR, Fowden AL & Giussani DA. (2002). Effects of prevailing hypoxaemia, acidaemia or hypoglycaemia upon the cardiovascular, endocrine and metabolic responses to acute hypoxaemia in the ovine fetus. *J*

Physiol 540, 351-366.

Gilbert RD. (1980). Control of fetal cardiac output during changes in blood volume. *Am J Physiol* 238, H80-86.

Jonker SS, Faber JJ, Anderson DF, Thornburg KL, Louey S & Giraud GD. (2006). Sequential growth of fetal sheep cardiac myocytes in response to simultaneous arterial and venous hypertension. *Am J Physiol Regul Integr Comp Physiol*.

Jonker SS, Giraud MK, Giraud GD, Chattergoon NN, Louey S, Davis LE, Faber JJ & Thornburg KL. (2010). Cardiomyocyte enlargement, proliferation and maturation during chronic fetal anaemia in sheep. *Exp Physiol* 95, 131-139.

Jonker SS, Louey S, Giraud GD, Thornburg KL & Faber JJ. (2015). Timing of cardiomyocyte growth, maturation, and attrition in perinatal sheep. *Faseb j* 29, 4346-4357.

Louey S, Jonker SS, Giraud GD & Thornburg KL. (2007). Placental insufficiency decreases cell cycle activity and terminal maturation in fetal sheep cardiomyocytes. *J Physiol* 580, 639-648.

Masoumy EP, Sawyer AA, Sharma S, Patel JA, Gordon PMK, Regnault TRH, Matuszewski B, Weintraub NL, Richardson B, Thompson JA & Stansfield BK. (2018). The lifelong impact of fetal growth restriction on cardiac development. *Pediatr Res* 84, 537-544.

Miller SL, Supramaniam VG, Jenkin G, Walker DW & Wallace EM. (2009). Cardiovascular responses to maternal betamethasone administration in the intrauterine growth-restricted ovine fetus. *Am J Obstet Gynecol* 201, 613 e611-618.

Morrison JL. (2008). Sheep models of intrauterine growth restriction: fetal adaptations and consequences. *Clin Exp Pharmacol Physiol* 35, 730-743.

Morrison JL, Botting KJ, Dyer JL, Williams SJ, Thornburg KL & McMillen IC. (2007). Restriction of placental function alters heart development in the sheep fetus. *Am J Physiol RIC* 293, R306-313.

Oh C, Dong Y, Liu H & Thompson LP. (2008). Intrauterine hypoxia upregulates proinflammatory cytokines and matrix metalloproteinases in fetal guinea pig hearts. *Am J Obstet Gynecol* 199, 717-726.

Poudel R, McMillen IC, Dunn SL, Zhang S & Morrison JL. (2015). Impact of chronic hypoxemia on blood flow to the brain, heart, and adrenal gland in the late-gestation IUGR sheep fetus. *Am J Physiol Regul Integr Comp Physiol* 308, R151-162.

Rueda-Clausen CF, Dolinsky VW, Morton JS, Proctor SD, Dyck JR & Davidge ST. (2011). Hypoxia-induced intrauterine growth restriction increases the susceptibility of rats to high-fat diet-induced metabolic syndrome. *Diabetes* 60, 507-516.

Rueda-Clausen CF, Morton JS & Davidge ST. (2009). Effects of hypoxia-induced intrauterine growth restriction on cardiopulmonary structure and function during adulthood. *Cardiovasc Res* 81, 713-722.

Ruijtenbeek K, le Noble FA, Janssen GM, Kessels CG, Fazzi GE, Blanco CE & De Mey JG. (2000). Chronic hypoxia stimulates periarterial sympathetic nerve development in chicken embryo. *Circ* 102, 2892-2897.

Teitel DF, Dalinghaus M, Cassidy SC, Payne BD & Rudolph AM. (1991). In utero ventilation augments the left ventricular response to isoproterenol and volume loading in fetal sheep. *Pediatr Res* 29, 466-472.

Thompson JA, Gimbel SA, Richardson BS, Gagnon R & Regnault TR. (2011a). The effect of intermittent umbilical cord occlusion on elastin composition in the ovine fetus. *Reprod Sci* 18, 990-997.

Thompson JA, Gros R, Richardson BS, Piorkowska K & Regnault TR. (2011b). Central stiffening in adulthood linked to aberrant aortic remodeling under suboptimal intrauterine conditions. *Am J Physiol Regul Integr Comp Physiol* 301, R1731-1737.

Thompson JA, Piorkowska K, Gagnon R, Richardson BS & Regnault TR. (2013). Increased collagen deposition in the heart of chronically hypoxic ovine fetuses. *J Dev Orig Health Dis* 4, 470-478.

Thompson JA, Richardson BS, Gagnon R & Regnault TR. (2011c). Chronic intrauterine hypoxia interferes with aortic development in the late gestation ovine fetus. *J Physiol* 589, 3319-3332.

Thompson JA, Sarr O, Piorkowska K, Gros R & Regnault TR. (2014). Low birth weight followed by postnatal over-nutrition in the guinea pig exposes a predominant player in the development of vascular dysfunction. *J Physiol* 592, 5429-5443.

Thompson L, Dong Y & Evans L. (2008). Title: Chronic Hypoxia Increases Inducible NOS-derived Nitric Oxide in Fetal Guinea Pig Hearts. *Pediatr Res*.

Thompson LP, Aguan K, Pinkas G & Weiner CP. (2000a). Chronic hypoxia increases the NO contribution of acetylcholine vasodilation of the fetal guinea pig heart. *Am J Physiol* 279, R1813-1820.

- Thompson LP & Dong Y. (2005). Chronic hypoxia decreases endothelial nitric oxide synthase protein expression in fetal guinea pig hearts. *J Soc Gynecol Investig* 12, 388-395.
- Thompson LP, Liu H, Evans L & Mong JA. (2011d). Prenatal nicotine increases matrix metalloproteinase 2 (MMP-2) expression in fetal guinea pig hearts. *Reprod Sci* 18, 1103-1110.
- Thompson LP, Pinkas G & Weiner CP. (2000b). Chronic 17beta-estradiol replacement increases nitric oxide-mediated vasodilation of guinea pig coronary microcirculation. *Circulation* 102, 445-451.
- Van Hare GF, Hawkins JA, Schmidt KG & Rudolph AM. (1990). The effects of increasing mean arterial pressure on left ventricular output in newborn lambs. *Circ Res* 67, 78-83.
- Wang KC, Botting KJ, Padhee M, Zhang S, McMillen IC, Suter CM, Brooks DA & Morrison JL. (2012a). Early origins of heart disease: low birth weight and the role of the insulin-like growth factor system in cardiac hypertrophy. *Clin Exp Pharmacol Physiol* 39, 958-964.
- Wang KC, Botting KJ, Zhang S, McMillen IC, Brooks DA & Morrison JL. (2017). Akt signaling as a mediator of cardiac adaptation to low birth weight. *J Endocrinol* 233, R81-R94.
- Wang KC, Brooks DA, Botting KJ & Morrison JL. (2012b). IGF-2R-mediated signaling results in hypertrophy of cultured cardiomyocytes from fetal sheep. *Biol Reprod* 86, 183.
- Wang KC, Brooks DA, Summers-Pearce B, Bobrovskaya L, Tosh DN, Duffield JA, Botting KJ, Zhang S, McMillen IC & Morrison JL. (2015a). Low birth weight activates the renin-angiotensin system, but limits cardiac angiogenesis in early postnatal life. *American journal of physiology Regulatory, integrative and comparative physiology* 3, pii: e12270.
- Wang KC, Brooks DA, Thornburg KL & Morrison JL. (2012c). Activation of IGF-2R stimulates cardiomyocyte hypertrophy in the late gestation sheep fetus. *J Physiol* 590, 5425-5437.
- Wang KC, Lim CH, McMillen IC, Duffield JA, Brooks DA & Morrison JL. (2013). Alteration of cardiac glucose metabolism in association to low birth weight: experimental evidence in lambs with left ventricular hypertrophy. *Metabolism* 62, 1662-1672.
- Wang KC, Tosh DN, Zhang S, McMillen IC, Duffield JA, Brooks DA & Morrison JL. (2015b). IGF-2R-Galphaq signaling and cardiac hypertrophy in the low-birth-weight lamb. *Am J Physiol RIC* 308, R627-635.
- Wang KC, Zhang L, McMillen IC, Botting KJ, Duffield JA, Zhang S, Suter CM, Brooks DA & Morrison JL. (2011). Fetal growth restriction and the programming of heart growth and cardiac IGF-2 expression in the lamb. *J Physiol* 589, 4709-4722.
- Xue Q & Zhang L. (2009). Prenatal hypoxia causes a sex-dependent increase in heart susceptibility to ischemia and reperfusion injury in adult male offspring: role of protein kinase C epsilon. *The Journal of pharmacology and experimental therapeutics* 330, 624-632.
- Zhu MY, Milligan N, Keating S, Windrim R, Keunen J, Thakur V, Ohman A, Portnoy S, Sled JG, Kelly E, Yoo SJ, Gross-Wortmann L, Jaeggi E, Macgowan CK, Kingdom JC & Seed M. (2016). The hemodynamics of late-onset intrauterine growth restriction by MRI. *Am J Obstet Gynecol* 214, 367.e361-367.e317.

REFeree COMMENTS

Referee #2:

This timely and comprehensive review by Azman and colleagues highlights an understudied but crucial topic in fetal growth restriction around the timing of cardiopulmonary transition. The main focus of the review being prospective clinical studies which are summarised nicely in tables covering changes in both cardiac structure and functional responses to FGR and SGA fetuses. The review particularly highlights the lack of preclinical studies targeting the cardiopulmonary transition in growth restricted fetuses, a highly important area of future study. The review is generally well written but could benefit from an abstract or summary figure to summarise the overall trends observed in all of these studies. Below I have some minor corrections/suggestions to improve the manuscript.

Ref at line 86 for termed 'brain sparing response'

Line 87 Though there is preferential distribution to the spared organs, they still generally receive less substrate delivery than normal pregnancies. So is not strictly 'maintained'.

Please include some references at line 102 for the heart outcomes stated.

Though there is much discussion about gross ventricular changes, not much is included on the cellular level. I realise this is mostly focussed on clinical/human data, but there is evidence of changes in cardiomyocyte size and endowment have been reported in FGR animal studies (PMID: 22930271, PMID: 29561647), this has not been included in the manuscript other than what occurs normally in adult cardiomyocytes (such in figure 1).

Line 167 - is this referring to the preterm FGR infants? Please indicate in the manuscript.

Line 236 - This sentence does not read clearly. I think an extra word has been included that does not need to be or needs rephrasing.

Line 249 - timing is important but the variability can also be explained by the additional factor of severity of FGR as the authors have explained in the previous section. As the studies included in table 2 also use differing definitions for SGA/FGR.

Could the authors include PVR in table 2? Eg. Seghal 2013 does not include this information, but is included for Suciú 2023.

END OF COMMENTS

Response to Reviewer Comments

Manuscript ID: JP-TR-2025-289441

Manuscript Title: Compromised Cardiopulmonary Transition in Fetal Growth Restricted and Small for Gestational Age Neonates

Note on Authorship Change

We would like to inform the editorial team that an additional author, Suzanne L. Miller, has been added to the revised manuscript. This change reflects their significant intellectual contribution during the revision process, particularly in refining the interpretation of results in the context of FGR and brain sparing. Prof. Miller is a recognised expert in the field of FGR, and their input was instrumental in strengthening the conceptual framing and clinical relevance of the manuscript. All authors have reviewed and approved the final version, with the completed change of authorship form included with the revised manuscript.

Note on Line Numbering

Please note that all line numbers referenced in our responses correspond to the tracked changes PDF version of the manuscript, not the clean version. This was done to ensure clarity in referencing specific edits and revisions.

EDITOR COMMENTS

Reviewing Editor:

This review summarises the impact of FGR on the cardiopulmonary transition at birth. This is an important area of study and thus deserves a comprehensive review with reference to major works in the field over decades.

The authors use FGR/SGA throughout the review suggesting that these terms are synonymous. They are not. SGA is a normal baby that was always meant to be a small baby. A FGR baby is one that should have been bigger but has not reached its genetic potential for growth, depending on the cause for not reaching its genetic growth potential, that baby may have undergone adaptations to survive. One of these that is used clinically to distinguish between FGR and SGA is Doppler. The problem is that it is difficult to distinguish between FGR and SGA, not that they are both pathological. It is very helpful that the table indicates whether the original studies classified the babies as SGA or FGR.

Thank you for this comment. We strongly agree with the reviewer that it is critical that FGR and SGA are considered separately. We had included a statement in the first paragraph to guide the reader as to the clinical differences between FGR and SGA:

P4 lines 65–72

“Infants born small for gestational age (SGA) are defined by an estimated fetal weight below the 10th percentile for gestational age, whereas fetal growth restriction (FGR) additionally requires the confirmation of abnormal Doppler findings indicative of placental pathology, such as increased pulsatility index or absent/reversed end-diastolic flow in the umbilical artery (Figueras & Gratacós, 2014; Lees et al., 2020). Although a proportion of SGA infants may be

constitutionally small, it is important to recognise that up to 50% of FGR cases remain undetected before birth, resulting in a population of SGA infants potentially having an unrecognised or subclinical pathology (Ernst et al., 2017; Lappen & Myers, 2017)."

We had hoped this statement would make it clear that these were separate populations. Further we attempted to signpost the fact that both FGR and SGA would be used by stating, *"This review will include studies describing both FGR or SGA as pathologies contributing to altered cardiovascular function in the perinatal period."* (original manuscript – P3 line 72). We concede that the use of pathologies in that sentence could confuse, and have therefore changed this sentence to:

ADDED

P4–5 lines 93–95: "This review will include studies describing both FGR and SGA and examine perinatal cardiovascular outcomes related to each condition."

The inclusion of both SGA and FGR is important because, before the Delphi consensus, SGA and FGR were often misrepresented. Thus, much of this earlier work is likely relevant to FGR, even if there was mislabelling (indeed, even work that labelled infants as FGR/IUGR may have indeed been studying SGA infants). It is also reasonable to include SGA due to this latter point, meaning that there is less information available regarding the transition at birth in pure FGR populations.

Given the importance of ensuring the distinction between the two we have added a statement around the use of both terminologies throughout the documents, as well as supporting statements as to why.

ADDED

P4 lines 72–74

"Thus, it is critical that FGR and SGA infants are considered separately, given that the former is representative of a pathological condition and the latter reflects constitutionally small individuals."

Line 66-70 – is there original work that can be cited here as well?

Thank you for this suggestion; we have now added original studies describing the fetal cardiac remodelling following exposure to intrauterine insults in place of the review by Crispi et al. We have also described examples of different intrauterine insults that may result in fetal cardiac remodelling.

ADDED

P4 lines 82–87

"The heart is central to the fetal adaptive response to intrauterine insults (e.g. toxin exposure, twin-twin transfusion syndrome, preeclampsia and FGR) and remodels in response to changes in preload and afterload to maintain optimal oxygenation of critical developing organs such as the brain (Karatzas et al., 2002; Rychik et al., 2007; García-Otero, 2016; Rodríguez-Lopez et al., 2017; Youssef et al., 2020)."

Line 70-71 - 'at birth' is used twice.

The first instance of 'at birth' has now been removed.

Line 79 - The term hypoxia is used here; however, the sentence is referring to hypoxemia. When the ewe breaths air that is low in oxygen, she is in a hypoxic environment. When blood oxygen is low, hypoxemia is present.

Thank you, we have now replaced 'fetal hypoxia' with 'fetal hypoxemia' and accordingly replaced any instances of 'chronic hypoxia' to 'hypoxemia' when referring to outcomes of placental insufficiency with maternal normoxia, thus differentiating them from maternal hypoxia-induced outcomes.

Line 89 - FGR is associated with brainsparing, but there is not always an increase in blood flow to the brain (Miller et al., 2009; Poudel et al., 2015; Zhu et al., 2016; Darby et al., 2024). However, there may be a distribution of regional blood flow within the brain that supports clinical findings of increased blood flow velocity in the MCA.

We appreciate the fact that FGR does not always result in increased blood flow and have reworded this to "...increase or maintain blood flow to critical organs". We have also added a sentence to discuss the regional differences in cerebral blood flow.

ADDED

P5, lines 109–116

"The brain sparing response occurs in an attempt to maintain oxygen delivery to vital organs by altering the perfusion of various vascular beds, therefore increasing or maintaining blood flow to critical organs, such as the brain and adrenal glands (Giussani, 2016). This occurs at the expense of less critical organs, including the lungs and gastrointestinal tract (Giussani, 2016). While FGR is associated with the brain sparing response, this does not consistently translate to an increase in overall cerebral blood flow (Miller et al., 2009; Poudel et al., 2009; Zhu et al., 2016; Darby et al., 2024). Instead, the redistribution of regional cerebral perfusion may explain the elevated middle cerebral artery blood flow velocities observed clinically (Eixarch et al., 2008)."

Line 97 - Studies have been performed to understand the impact of chronic hypoxia on the brainstem. These should be cited (Adams et al., 2001).

We have now incorporated the range of brainstem outcomes observed in preclinical studies of chronic hypoxemia.

ADDED

P5–6, lines 128–134

"Preclinical studies have investigated the effects of chronic hypoxemia on the brainstem, reporting conflicting outcomes including minimal structural impact (Thordstein & Hedner, 1992; Adams et al., 2001; Tolcos et al., 2003), reduced brainstem blood flow (Miller et al., 2009) and dysregulated monoaminergic and neuroglial development (Tolcos & Rees, 1997; Ahmadzadeh et al., 2023a). Regardless, it is possible that neuropathology in the brainstem underlies the poor cardiorespiratory outcomes of FGR infants after birth (Ahmadzadeh et al., 2023b)."

Line 101-102 - References to the impact of FGR on the fetal heart should be used to support this statement. These may be clinical or preclinical findings in a range of species but sheep and guinea pig may be most relevant.

Thank you, we have now added references to preclinical studies describing the impact of FGR on the fetal heart.

ADDED

P6, lines 137–142

“Chronic brain sparing, maintained in the presence of hypoxemia, is also known to cause maladaptive morphological adaptations to the heart, resulting in cardiac remodelling (Pérez-Cruz et al., 2015; Rodríguez-López et al., 2017), disrupted cardiomyocyte maturation (Bubb et al., 2007; Louey et al., 2007; Jonker et al., 2018) and vascular dysfunction (Thompson et al., 2011a; Sehgal et al., 2013; Thompson et al., 2014), ultimately compromising cardiovascular function.”

Line 120 - This was shown much earlier than 2016. Please cite original work (Gilbert, 1980; Van Hare et al., 1990; Teitel et al., 1991).

We have now replaced the 2016 reference with the original papers listed above.

It is generally preferred to use 'that' rather than 'which'. However, when 'which' is used, it should be preceded by a comma except when used as 'by which', 'in which', etc.

Thank you for these corrections. All instances of “which” are either now preceded by a comma or changed to “that” where appropriate.

Others have previously suggested that timing, duration and severity of the causes of FGR impacts how the heart and other organs adapt to FGR (Morrison, 2008; Darby et al., 2020b).

The authors appreciate this suggestion and agree that it is important to highlight that these factors all affect the FGR heart and other developing organs in different manners.

ADDED

P6 lines 143–145

“Importantly, the timing, duration and severity of the underlying causes of FGR each uniquely influence the developmental trajectory of the heart and other organs, leading to variable structural and functional outcomes (Morrison, 2008; Darby et al., 2020, Dudink et al., 2025).”

Line 193 - The sentence that ends on this line could be supported by a range of references.

We have now supplemented this sentence with its appropriate references.

Line 249 - The use of the term 'varying' is quite vague. It is not clear if this refers to the fact that normal cardiac maturation occurs during this time (Jonker et al., 2006; Jonker et al., 2010; Jonker et al., 2015) or if FGR can impact that maturation depending upon the timing, duration and severity of the cause of the FGR (Thompson et al., 2000a; Thompson et al., 2000b; Thompson & Dong, 2005; Bubb et al., 2007; Louey et al., 2007; Morrison et al., 2007; Oh et al., 2008; Thompson et al., 2008; Thompson et al., 2011d; Wang et al., 2011; Evans et al., 2012a; Evans et al., 2012b; Wang et al., 2012a; Wang et al., 2012b; Wang et al., 2012c; Al-Hasan et al., 2013; Thompson et al., 2013; Wang et al., 2013; Al-Hasan et al., 2014; Botting et al., 2014; Wang et al., 2015a; Wang et al., 2015b; Wang et al., 2017; Botting et al., 2018; Darby et al., 2018; Masoumy et al., 2018; Dimasi et al., 2021; Dimasi et al., 2023; Dimasi et al., 2024a; Dimasi et al., 2024b).

We agree that the use of 'varying' in the initial sentence does not reflect the different roles and thus impact of normal cardiac maturation and timing/severity of FGR. We have now altered these sentences (*P11, lines 292–299*) to clarify the role of both normal maturation and how this may be influenced by FGR.

ADDED

P11, lines 292–299

“The variability in findings across these studies is likely due to the heterogeneity in gestational ages among study populations, ranging from <28 weeks to ≥39 weeks, during which substantial cardiovascular maturation normally occurs (Jonker et al., 2007; Jonker et al., 2010; Jonker et al., 2015). This variation in developmental stage may also interact with the timing, duration and severity of FGR, which are all known to influence cardiovascular function at the time of assessment (Thompson et al., 2000; Bubb et al., 2007; Louey et al., 2007; Morrison et al., 2007; Wang et al., 2011; Botting et al., 2014; Dimasi et al., 2021; Dimasi et al., 2023a).”

Line 340 - this sentence could be supported with references.

Thank you. We have expanded this section regarding the paucity of preclinical data investigating the immediate postnatal period and have therefore supported it with several of the references suggested from your comments.

Line 341 - The authors introduce a preclinical model of FGR in sheep. There are several sheep models of FGR. It would be useful to describe this model e.g. is there placental insufficiency? FGR? Chronic hypoxemia? Brainsparing?.

We agree that a more comprehensive description of this model would be useful and have now specified the effects of this FGR model on the fetus. We have also included similar descriptions in other instances where a preclinical FGR model is mentioned (maternal hypoxia, carunclectomy, etc.)

ADDED

P14, lines 395–398

“One study assessed lambs delivered at 0.85 gestation, where FGR was induced by single umbilical artery ligation (SUAL) at 0.7 gestation, resulting in placental insufficiency, chronic hypoxemia and asymmetrical growth restriction (i.e. brain sparing) (Polglase et al., 2016; Sutherland et al., 2024).”

Line 349 - There are many studies of the impact of FGR on aortic stiffness (Thompson et al., 2011a; Thompson et al., 2011b; Thompson et al., 2011c; Thompson et al., 2014).

Thank you for this comment; we appreciate the importance of other work describing aortic stiffness in FGR especially given the increased risk of hypertension in later life. However, in this section we are aiming to specify studies conducted immediately after birth to follow the discussion of postnatal echocardiographic changes in clinical studies, whereas the referenced studies are either conducted before birth (Thompson 2011a, Thompson 2011b, Thompson 2011c) or during postnatal ages equivalent to adolescence (Thompson 2014). Regardless, as we have expanded the discussion on preclinical data, these studies have been added into the paragraphs on vascular dysfunction (*P15, lines 430–457*).

Line 350 - Please reword this sentence to more clearly indicate that FGR was induced by “...” in rats.

We apologise for the vagueness of this sentence. As the previous sentence had described the mode of FGR induction in the same study, we omitted it in the sentence referred to. We have now altered lines 462–465 so that the referenced sentence is less ambiguous.

ADDED

P14, lines 401–404

“In FGR rat pups induced by chronic maternal hypoxia (11.5% O₂) from 0.71 gestation until term, aortic stiffness and decreased heart rate was observed on the first day of life despite echocardiographic parameters being comparable to AG offspring (Kumar et al., 2020).”

Line 357 - It would be useful to more clearly articulate the gaps in knowledge. This could be summarised in a figure.

Thank you, we have now summarised this in Figure 2, which is consistent with the expansion of the “Preclinical studies” section (P13–16, lines 381–488).

Line 359 - ... impairments to persist...

This line now reads as “suggested that these impairments may persist” for better clarity.

It may well be that biomarkers will play an important role in improving the ability to detect FGR. However, the discussion on the challenges of distinguishing between SGA and FGR and the clinical benefit of this should be outlined in the review prior to this section. The mechanistic underpinning of the discussed biomarkers should also be clear earlier in the piece.

Thank you for the comment. We have added some additional discussion around the need for biomarkers throughout the review. Please see additions below.

ADDED

P4 lines 75–78

“Further, given the current low rate of detection of true FGR, the ability to accurately distinguish between FGR and SGA populations remains a clinical challenge, underscoring the need for mechanistically informed biomarkers that can add clarity to the clinical interpretation of Doppler scans.”

P7 lines 178–179

“The mechanistic pathways contributing to cardiovascular sequelae in FGR may also offer a foundation for developing biomarkers to improve the diagnosis of FGR.”

It is true that some fetuses are more vulnerable to a second hit. The concept was outlined some time ago and likely applies to many scenarios, including this one. It would be appropriate to acknowledge this (Rueda-Clausen et al., 2009; Xue & Zhang, 2009; Rueda-Clausen et al., 2011).

We appreciate the added nuance that the reviewer has suggested with this comment. We have now incorporated these studies into the paragraph (P18, lines 552–560) along with a concluding sentence (P18, lines 564–565) to highlight the vulnerability of the FGR infant to the second hit.

ADDED

P18, lines 552–560

“This concept aligns with the “double hit” hypothesis, where exposure to in utero chronic hypoxemia programs the fetus with latent vulnerabilities that may not manifest until a secondary challenge occurs. Preclinical studies have shown that FGR offspring are more susceptible to adverse cardiopulmonary outcomes with aging, specifically increased myocardial injury following ischemia-reperfusion (Rueda-Clausen et al., 2009) and metabolic dysfunction in response to a high-fat diet (Rueda-Clausen et al., 2011a). These programmed susceptibilities are mediated by disrupted signalling pathways, such as PKCε repression and impaired insulin signalling, and may occur in a sex-specific manner (Xue & Zhang, 2009; Rueda-Clausen et al., 2011a).”

Additionally, the start of the next paragraph now highlights the focus on in vivo physiological insults, rather than the broader metabolic or ischemia-reperfusion insults.

ADDED

P18, lines 567–570

“Much of our current understanding on the ability of the FGR/SGA neonate to respond to secondary physiological insults, such as birth asphyxia or cardiovascular instability, is derived from preclinical studies investigating in vivo haemodynamic responses in sheep.”

Line 402-418 - There are other studies that have investigated the impact of additional insults in FGR and chronic hypoxemia (Gardner et al., 2002; Darby et al., 2020a; Darby et al., 2024).

Thank you for this suggestion. We have now incorporated these studies into the discussion of secondary insults via maternal hypoxia.

Line 425 -427 - There are papers that have studied the impact of FGR on regulation of blood pressure in the fetus (Ruijtenbeek et al., 2000; Danielson et al., 2005; Dyer et al., 2009).

Thank you. We have now added the studies by Danielson, Dyer and Darby into the discussion of blood pressure manipulation as a secondary insult. The Ruijtenbeek study does not appear to incorporate a secondary hit on top of chronic hypoxaemia and has therefore not been included.

References

Adams MB, Brown RE, Gibson C, Coulter CL & McMillen IC. (2001). Tyrosine hydroxylase protein content in the medulla oblongata of the foetal sheep brain increases in response to acute but not chronic hypoxia. *Neurosci Lett* 316, 63-66.

Al-Hasan YM, Evans LC, Pinkas GA, Dabkowski ER, Stanley WC & Thompson LP. (2013). Chronic hypoxia impairs cytochrome oxidase activity via oxidative stress in selected fetal Guinea pig organs. *Reprod Sci* 20, 299-307.

Al-Hasan YM, Pinkas GA & Thompson LP. (2014). Prenatal Hypoxia Reduces Mitochondrial Protein Levels and Cytochrome c Oxidase Activity in Offspring Guinea Pig Hearts. *Reprod Sci* 21, 883-891.

Botting KJ, Loke XY, Zhang S, Andersen JB, Nyengaard JR & Morrison JL. (2018). IUGR decreases cardiomyocyte endowment and alters cardiac metabolism in a sex and cause of IUGR specific manner. *Am J Physiol Regul Integr Comp Physiol*.

Botting KJ, McMillen IC, Forbes H, Nyengaard JR & Morrison JL. (2014). Chronic hypoxemia in late gestation decreases cardiomyocyte number but does not change expression of hypoxia-responsive genes. *J Am Heart Assoc* 3, pii: e000531.

Bubb KJ, Cock ML, Black MJ, Dodic M, Boon WM, Parkington HC, Harding R & Tare M. (2007). Intrauterine growth restriction delays cardiomyocyte maturation and alters coronary artery function in the fetal sheep. *J Physiol* 578, 871- 881.

Danielson L, McMillen IC, Dyer JL & Morrison JL. (2005). Restriction of placental growth results in greater hypotensive response to α -adrenergic blockade in fetal sheep during late gestation. *J Physiol* 563, 611-620.

Darby JRT, McMillen IC & Morrison JL. (2018). Maternal undernutrition in late gestation increases IGF2 signalling molecules and collagen deposition in the right ventricle of the fetal sheep heart. *J Physiol* 596, 2345-2358.

Darby JRT, Saini BS, Holman SL, Hammond SJ, Perumal SR, Macgowan CK, Seed M & Morrison JL. (2024). Acute- on-chronic: using magnetic resonance imaging to disentangle the haemodynamic responses to acute and chronic fetal hypoxaemia. *Front Med (Lausanne)* 11, 1340012.

Darby JRT, Varcoe TJ, Holman SL, McMillen IC & Morrison JL. (2020a). The reliance on α -adrenergic receptor stimuli for blood pressure regulation in the chronically hypoxaemic fetus is not dependent on post-ganglionic activation. *J Physiol* 599, 1307-1318.

Darby JRT, Varcoe TJ, Orgeig S & Morrison JL. (2020b). Cardiorespiratory consequences of intrauterine growth restriction: Influence of timing, severity and duration of hypoxaemia. *Theriogenology* 150, 84-95.

Dimasi CG, Darby JRT, Cho SKS, Saini BS, Holman SL, Meakin AS, Wiese MD, Macgowan CK, Seed M & Morrison JL. (2024a). Reduced in utero substrate supply decreases mitochondrial abundance and alters the expression of metabolic signalling molecules in the fetal sheep heart. *J Physiol* 602, 5901-5922.

Dimasi CG, Darby JRT, Holman SL, Quinn M, Meakin AS, Seed M, Wiese MD & Morrison JL. (2024b). Cardiac growth patterns and metabolism before and after birth in swine: Role of miR in proliferation, hypertrophy and metabolism. *J Mol Cell Cardiol Plus* 9, 100084.

Dimasi CG, Darby JRT & Morrison JL. (2023). A change of heart: understanding the mechanisms regulating cardiac proliferation and metabolism before and after birth. *J Physiol* 601, 1319-1341.

Dimasi CG, Lazniewska J, Plush SE, Saini BS, Holman SL, Cho SKS, Wiese MD, Sorvina A, Macgowan CK, Seed M, Brooks DA, Morrison JL & Darby JRT. (2021). Redox ratio in the left

ventricle of the growth restricted fetus is positively correlated with cardiac output. *J Biophotonics*, e202100157.

Dyer JL, McMillen IC, Warnes KE & Morrison JL. (2009). No Evidence for an Enhanced Role of Endothelial Nitric Oxide in the Maintenance of Arterial Blood Pressure in the IUGR Sheep Fetus. *Placenta* 30, 705-710.

Evans LC, Liu H, Pinkas GA & Thompson LP. (2012a). Chronic hypoxia increases peroxynitrite, MMP9 expression, and collagen accumulation in fetal guinea pig hearts. *Pediatr Res* 71, 25-31.

Evans LC, Liu H & Thompson LP. (2012b). Differential effect of intrauterine hypoxia on caspase 3 and DNA fragmentation in fetal guinea pig hearts and brains. *Reprod Sci* 19, 298-305.

Gardner DS, Fletcher AJ, Bloomfield MR, Fowden AL & Giussani DA. (2002). Effects of prevailing hypoxaemia, acidaemia or hypoglycaemia upon the cardiovascular, endocrine and metabolic responses to acute hypoxaemia in the ovine fetus. *J Physiol* 540, 351-366.

Gilbert RD. (1980). Control of fetal cardiac output during changes in blood volume. *Am J Physiol* 238, H80-86.

Jonker SS, Faber JJ, Anderson DF, Thornburg KL, Louey S & Giraud GD. (2006). Sequential growth of fetal sheep cardiac myocytes in response to simultaneous arterial and venous hypertension. *Am J Physiol Regul Integr Comp Physiol*.

Jonker SS, Giraud MK, Giraud GD, Chattergoon NN, Louey S, Davis LE, Faber JJ & Thornburg KL. (2010). Cardiomyocyte enlargement, proliferation and maturation during chronic fetal anaemia in sheep. *Exp Physiol* 95, 131- 139.

Jonker SS, Louey S, Giraud GD, Thornburg KL & Faber JJ. (2015). Timing of cardiomyocyte growth, maturation, and attrition in perinatal sheep. *Faseb j* 29, 4346-4357.

Louey S, Jonker SS, Giraud GD & Thornburg KL. (2007). Placental insufficiency decreases cell cycle activity and terminal maturation in fetal sheep cardiomyocytes. *J Physiol* 580, 639-648.

Masoumy EP, Sawyer AA, Sharma S, Patel JA, Gordon PMK, Regnault TRH, Matuszewski B, Weintraub NL, Richardson B, Thompson JA & Stansfield BK. (2018). The lifelong impact of fetal growth restriction on cardiac development. *Pediatr Res* 84, 537-544.

Miller SL, Supramaniam VG, Jenkin G, Walker DW & Wallace EM. (2009). Cardiovascular responses to maternal betamethasone administration in the intrauterine growth-restricted ovine fetus. *Am J Obstet Gynecol* 201, 613 e611- 618.

Morrison JL. (2008). Sheep models of intrauterine growth restriction: fetal adaptations and consequences. *Clin Exp Pharmacol Physiol* 35, 730-743.

Morrison JL, Botting KJ, Dyer JL, Williams SJ, Thornburg KL & McMillen IC. (2007). Restriction of placental function alters heart development in the sheep fetus. *Am J Physiol RIC* 293, R306-313.

Oh C, Dong Y, Liu H & Thompson LP. (2008). Intrauterine hypoxia upregulates proinflammatory cytokines and matrix metalloproteinases in fetal guinea pig hearts. *Am J Obstet Gynecol* 199, 78.e71-76.

Poudel R, McMillen IC, Dunn SL, Zhang S & Morrison JL. (2015). Impact of chronic hypoxemia on blood flow to the brain, heart, and adrenal gland in the late-gestation IUGR sheep fetus. *Am J Physiol Regul Integr Comp Physiol* 308, R151-162.

Rueda-Clausen CF, Dolinsky VW, Morton JS, Proctor SD, Dyck JR & Davidge ST. (2011). Hypoxia-induced intrauterine growth restriction increases the susceptibility of rats to high-fat diet-induced metabolic syndrome. *Diabetes* 60, 507-516.

Rueda-Clausen CF, Morton JS & Davidge ST. (2009). Effects of hypoxia-induced intrauterine growth restriction on cardiopulmonary structure and function during adulthood. *Cardiovasc Res* 81, 713-722.

Ruijtenbeek K, le Noble FA, Janssen GM, Kessels CG, Fazzi GE, Blanco CE & De Mey JG. (2000). Chronic hypoxia stimulates periarterial sympathetic nerve development in chicken embryo. *Circ* 102, 2892-2897.

Teitel DF, Dalinghaus M, Cassidy SC, Payne BD & Rudolph AM. (1991). In utero ventilation augments the left ventricular response to isoproterenol and volume loading in fetal sheep. *Pediatr Res* 29, 466-472.

Thompson JA, Gimbel SA, Richardson BS, Gagnon R & Regnault TR. (2011a). The effect of intermittent umbilical cord occlusion on elastin composition in the ovine fetus. *Reprod Sci* 18, 990-997.

Thompson JA, Gros R, Richardson BS, Piorkowska K & Regnault TR. (2011b). Central stiffening in adulthood linked to aberrant aortic remodeling under suboptimal intrauterine conditions. *Am J Physiol Regul Integr Comp Physiol* 301, R1731-1737.

Thompson JA, Piorkowska K, Gagnon R, Richardson BS & Regnault TR. (2013). Increased collagen deposition in the heart of chronically hypoxic ovine fetuses. *J Dev Orig Health Dis* 4, 470-478.

Thompson JA, Richardson BS, Gagnon R & Regnault TR. (2011c). Chronic intrauterine hypoxia interferes with aortic development in the late gestation ovine fetus. *J Physiol* 589, 3319-3332.

Thompson JA, Sarr O, Piorkowska K, Gros R & Regnault TR. (2014). Low birth weight followed by postnatal over-nutrition in the guinea pig exposes a predominant player in the development of vascular dysfunction. *J Physiol* 592, 5429-5443.

Thompson L, Dong Y & Evans L. (2008). Title: Chronic Hypoxia Increases Inducible NOS-derived Nitric Oxide in Fetal Guinea Pig Hearts. *Pediatr Res*.

Thompson LP, Aguan K, Pinkas G & Weiner CP. (2000a). Chronic hypoxia increases the NO contribution of acetylcholine vasodilation of the fetal guinea pig heart. *Am J Physiol* 279, R1813-1820.

Thompson LP & Dong Y. (2005). Chronic hypoxia decreases endothelial nitric oxide synthase protein expression in fetal guinea pig hearts. *J Soc Gynecol Investig* 12, 388-395.

Thompson LP, Liu H, Evans L & Mong JA. (2011d). Prenatal nicotine increases matrix metalloproteinase 2 (MMP-2) expression in fetal guinea pig hearts. *Reprod Sci* 18, 1103-1110.

Thompson LP, Pinkas G & Weiner CP. (2000b). Chronic 17beta-estradiol replacement increases nitric oxide-mediated vasodilation of guinea pig coronary microcirculation. *Circulation* 102, 445-451.

Van Hare GF, Hawkins JA, Schmidt KG & Rudolph AM. (1990). The effects of increasing mean arterial pressure on left ventricular output in newborn lambs. *Circ Res* 67, 78-83.

Wang KC, Botting KJ, Padhee M, Zhang S, McMillen IC, Suter CM, Brooks DA & Morrison JL. (2012a). Early origins of heart disease: low birth weight and the role of the insulin-like growth factor system in cardiac hypertrophy. *Clin Exp Pharmacol Physiol* 39, 958-964.

Wang KC, Botting KJ, Zhang S, McMillen IC, Brooks DA & Morrison JL. (2017). Akt signaling as a mediator of cardiac adaptation to low birth weight. *J Endocrinol* 233, R81-R94.

Wang KC, Brooks DA, Botting KJ & Morrison JL. (2012b). IGF-2R-mediated signaling results in hypertrophy of cultured cardiomyocytes from fetal sheep. *Biol Reprod* 86, 183.

Wang KC, Brooks DA, Summers-Pearce B, Bobrovskaya L, Tosh DN, Duffield JA, Botting KJ, Zhang S, McMillen IC & Morrison JL. (2015a). Low birth weight activates the renin-angiotensin system, but limits cardiac angiogenesis in early postnatal life. *American journal of physiology Regulatory, integrative and comparative physiology* 3, pii: e12270.

Wang KC, Brooks DA, Thornburg KL & Morrison JL. (2012c). Activation of IGF-2R stimulates cardiomyocyte hypertrophy in the late gestation sheep fetus. *J Physiol* 590, 5425-5437.

Wang KC, Lim CH, McMillen IC, Duffield JA, Brooks DA & Morrison JL. (2013). Alteration of cardiac glucose metabolism in association to low birth weight: experimental evidence in lambs with left ventricular hypertrophy. *Metabolism* 62, 1662-1672.

Wang KC, Tosh DN, Zhang S, McMillen IC, Duffield JA, Brooks DA & Morrison JL. (2015b). IGF-2R-Galphaq signaling and cardiac hypertrophy in the low-birth-weight lamb. *Am J Physiol RIC* 308, R627-635.

Wang KC, Zhang L, McMillen IC, Botting KJ, Duffield JA, Zhang S, Suter CM, Brooks DA & Morrison JL. (2011). Fetal growth restriction and the programming of heart growth and cardiac IGF-2 expression in the lamb. *J Physiol* 589, 4709-4722.

Xue Q & Zhang L. (2009). Prenatal hypoxia causes a sex-dependent increase in heart susceptibility to ischemia and reperfusion injury in adult male offspring: role of protein kinase C epsilon. *The Journal of pharmacology and experimental therapeutics* 330, 624-632.

Zhu MY, Milligan N, Keating S, Windrim R, Keunen J, Thakur V, Ohman A, Portnoy S, Sled JG, Kelly E, Yoo SJ, Gross-Wortmann L, Jaeggi E, Macgowan CK, Kingdom JC & Seed M. (2016). The hemodynamics of late-onset intrauterine growth restriction by MRI. *Am J Obstet Gynecol* 214, 367.e361-367.e317.

REFEREE COMMENTS

Referee #2:

This timely and comprehensive review by Azman and colleagues highlights an understudied but crucial topic in fetal growth restriction around the timing of cardiopulmonary transition. The main focus of the review being prospective clinical studies which are summarised nicely in tables covering changes in both cardiac structure and functional responses to FGR and SGA fetuses. The review particularly highlights the lack of preclinical studies targeting the cardiopulmonary transition in growth restricted fetuses, a highly important area of future study. The review is generally well written but could benefit from an abstract or summary figure to summarise the overall trends observed in all of these studies. Below I have some minor corrections/suggestions to improve the manuscript.

Thank you for this comment. We have now included an abstract figure with this review.

Ref at line 86 for termed 'brain sparing response'

Thank you, we have now added a reference for this statement.

Line 87 Though there is preferential distribution to the spared organs, they still generally receive less substrate delivery than normal pregnancies. So is not strictly 'maintained'.

We appreciate that the brain sparing response is only an attempt to maintain perfusion and have now specified this in the manuscript.

ADDED

P5, lines 109–111

“The brain sparing response occurs in an attempt to maintain oxygen delivery to vital organs by altering the perfusion of various vascular beds, therefore increasing or maintaining blood flow to critical organs, such as the brain and adrenal glands (Giussani, 2016).”

We agree the reduction in substrate delivery is an important distinction and have added a sentence stating this outcome.

ADDED

P5, lines 116–119

“Additionally, although blood flow is preferentially redistributed to critical organs following chronic hypoxemia, overall substrate delivery is reduced compared to normal pregnancies, resulting in reduced fetal substrate consumption relative to oxygen delivery (Cetin et al., 2020).”

Please include some references at line 102 for the heart outcomes stated.

Thank you, we have now added references for each of the heart outcomes stated.

Though there is much discussion about gross ventricular changes, not much is included on the cellular level. I realise this is mostly focussed on clinical/human data, but there is evidence of changes in cardiomyocyte size and endowment have been

reported in FGR animal studies (PMID: 22930271, PMID: 29561647), this has not been included in the manuscript other than what occurs normally in adult cardiomyocytes (such in figure 1).

Thank you for your comment. We agree that the manuscript lacked preclinical data, which is a result of few studies being conducted in the field and have attempted to signpost this in lines 382–385: *“There is a paucity of preclinical data regarding the effects of the cardiopulmonary transition at birth on the growth-restricted heart, with most studies reporting cardiovascular dysfunction either before birth or at much older postnatal ages equivalent to adolescent or adult cardiac development, with the current gap in knowledge summarised in Figure 2.”*

However, as the Reviewing Editor had similarly suggested the addition of a figure (Figure 2) to discuss the gaps in preclinical data, we have now expanded on this section (P13–16, lines 381–488). We hope that the expansion of this section has now clearly articulated the aspects of preclinical data that remains lacking in the field.

Line 167 - is this referring to the preterm FGR infants? Please indicate in the manuscript.

Thank you for this correction, we have now specified that this finding refers to preterm FGR infants.

ADDED

P7, lines 185–187

“On echocardiographic assessment, this spherical appearance reflects in a lower sphericity index in the postnatal period in preterm FGR infants (Sehgal et al., 2017).”

Line 236 - This sentence does not read clearly. I think an extra word has been included that does not need to be or needs rephrasing.

Thank you, we have now split this sentence into two to improve its clarity.

ADDED

P10–11, lines 279–283

“Similarly, Fouzas et al. reported that preterm AG infants showed an increase in LV SV between postnatal days 2 and 5 (Fouzas et al., 2014). This, along with a decrease in MPI and heart rate, suggests an overall improvement in global myocardial function in AG infants over time (Fouzas et al., 2014).”

Line 249 - timing is important but the variability can also be explained by the additional factor of severity of FGR as the authors have explained in the previous section. As the studies included in table 2 also use differing definitions for SGA/FGR.

We have now altered this sentence to reflect the influence of both timing and severity on cardiovascular outcomes, given that the Reviewing Editor had similarly commented on the ambiguity of this sentence.

ADDED

P11, lines 292–299

“The variability in findings across these studies is likely due to the heterogeneity in gestational ages among study populations, ranging from <28 weeks to ≥39 weeks, during which substantial cardiovascular maturation normally occurs (Jonker et al., 2007; Jonker et al., 2010;

Jonker et al., 2015). This variation in developmental stage may also interact with the timing, duration and severity of FGR, which are all known to influence cardiovascular function at the time of assessment (Thompson et al., 2000; Bubb et al., 2007; Louey et al., 2007; Morrison et al., 2007; Wang et al., 2011; Botting et al., 2014; Dimasi et al., 2021; Dimasi et al., 2023a)."

Could the authors include PVR in table 2? Eg. Sehgal 2013 does not include this information, but is included for Suciu 2023.

Thank you. We have now included measures of PVR in the Sehgal studies in Table 2.

Dear Ms Azman,

Re: JP-TR-2025-289441R1 "**Compromised Cardiopulmonary Transition in Fetal Growth Restricted and Small for Gestational Age Neonates**" by Zahrah Azman, Arvind Sehgal, Suzanne L Miller, Kristen J Bubb, Graeme R Polglase, and Beth J Allison

We are pleased to tell you that your paper has been accepted for publication in The Journal of Physiology.

Authors should note that it is too late at this point to offer corrections prior to proofing. Major corrections at proof stage, such as changes to figures, will be referred to the Editors for approval before they can be incorporated. Only minor changes, such as to style and consistency, should be made at proof stage. Changes that need to be made after proof stage will usually require a formal correction notice.

Yours sincerely,

Laura Bennet
Senior Editor
The Journal of Physiology

P.S. - You can help your research get the attention it deserves! Check out Wiley's free Promotion Guide for best-practice recommendations for promoting your work at www.wileyauthors.com/eeo/guide. You can learn more about Wiley Editing Services which offers professional video, design, and writing services to create shareable video abstracts, infographics, conference posters, lay summaries, and research news stories for your research at www.wileyauthors.com/eeo/promotion.

IMPORTANT NOTICE ABOUT OPEN ACCESS: To assist authors whose funding agencies mandate public access to published research findings sooner than 12 months after publication, The Journal of Physiology allows authors to pay an Open Access (OA) fee to have their papers made freely available immediately on publication.

You can check if your funder or institution has a Wiley Open Access Account here: <https://authorservices.wiley.com/author-resources/Journal-Authors/licensing-and-open-access/open-access/author-compliance-tool.html>.

EDITOR COMMENTS

Reviewing Editor:

Thank you for addressing the reviewer's comments. This is a lovely review.

REFeree COMMENTS

Referee #2:

Thank you for addressing my comments. The review has improved greatly. I have no further comments.